# High Precision 3D Printing for Micro to Nano Scale Biomedical and Electronic Devices

**DOI:** 10.3390/mi13040642

**Published:** 2022-04-18

**Authors:** Kirsty Muldoon, Yanhua Song, Zeeshan Ahmad, Xing Chen, Ming-Wei Chang

**Affiliations:** 1Nanotechnology and Integrated Bioengineering Centre, University of Ulster, Jordanstown Campus, Newtownabbey BT37 0QB, UK; muldoon-k5@ulster.ac.uk; 2Key Laboratory for Biomedical Engineering of Education Ministry of China, Zhejiang University, Hangzhou 310027, China; syanhua2015@163.com; 3Zhejiang Provincial Key Laboratory of Cardio-Cerebral Vascular Detection Technology and Medical Effectiveness Appraisal, Zhejiang University, Hangzhou 310027, China; 4School of Pharmacy, De Montfort University, Leicester LE1 9BH, UK; zahmad@dmu.ac.uk

**Keywords:** 3D printing, micro/nano scale printing, biomaterial, electronics

## Abstract

Three dimensional printing (3DP), or additive manufacturing, is an exponentially growing process in the fabrication of various technologies with applications in sectors such as electronics, biomedical, pharmaceutical and tissue engineering. Micro and nano scale printing is encouraging the innovation of the aforementioned sectors, due to the ability to control design, material and chemical properties at a highly precise level, which is advantageous in creating a high surface area to volume ratio and altering the overall products’ mechanical and physical properties. In this review, micro/-nano printing technology, mainly related to lithography, inkjet and electrohydrodynamic (EHD) printing and their biomedical and electronic applications will be discussed. The current limitations to micro/-nano printing methods will be examined, covering the difficulty in achieving controlled structures at the miniscule micro and nano scale required for specific applications.

## 1. Introduction

The 3D printing market was valued at $13.78 billion in 2020 and is expected to grow by 21% from 2021 to 2028 [1]. The sales market of 3D printers has seen 2.2 million shipments in 2021, which is expected to increase to 21.5 million by 2030. This increase is estimated to be a result of research and development, particularly within the medical and automotive sectors. Prototyping is the leading application in the 3D printing market, attributed to more than 55% of global revenue in 2021. This is likely due to prototyping achieving higher accuracy to develop high end products [2]. The popularity of the additive manufacturing process in recent years is due to the ability of altering design, rapid manufacturing and its cost effectiveness for production in industry. The technology integrates materials, structures and functions which are applied to many sectors of engineering [3]. The capability and specification of 3D printers are constantly evolving from the initial basic printer which only produced designs with one material, whereas multiple material extrusion is now possible and the range of materials suitable for printing is growing due to the evolvement of the printing machines and material science. The general process involves the creation of a model using CAD (computer aided design), which is then sliced according to the layer building process. The 3D printer deposits material in a three dimensional range (x, y and z) according to the printing process employed [4]. As printing processes progress, the precision scale has been minimized to the micro and nano level. This scale has an effect on the material properties, an example is gold in bulk form which has a high melting point of 1064 °C [5], whereas gold nanoparticles have a much lower melting point at approximately 23–25 °C [5,6]. Popular examples of micro-/nano scale printing that are well developed for precision fabrication are inkjet, electrohydrodynamic (EHD) and stereolithography (SLA); although they are revolutionary on such a small scale, some difficulties still occur within the process. Inkjet printing is typically limited to the sub-micrometer scale specifically having the ability to print in the range of 1–500 µm [7]. Electrohydrodynamic printing requires a highly controlled processing environment and the use of surfactant and additives to achieve complex structures [8]. Stereolithography can be problematic due to the limited strength of the cured resin and the small batch capabilities [9]. A widespread limitation to micro-/nano printing is the difficulty in executing 3D structures due to fabrication limits. Although direct printing techniques allow the deposition of minuscule amounts of material effectively, it is still a challenge to exploit the full potential of fabrication of 3D structures [10]. The desire for micro and nano structures to be incorporated with the 3DP bulk is because of the advanced properties and applications it can offer. Structures include micro-/nano coatings [11], filaments [12], tubes [13] and particles [14] etc., which offer special features, levels, dimensions and physical properties [15]. Reported applications implement micro-pits and nanotubes to enhance cell proliferation, surface hydrophilicity and bioactivity in biomedical applications [16]. In addition, wearable technology integrating micro/nano structures has been a growing sector in recent years, with many people possessing a wearable device globally. The printed structure is significant in the design of wearable technology, with studies showing the ability of structures acting as sensing layers with high sensitivity, selectivity and fast response time [17]. 

This review focuses on comprehensive discussions of micro-/nano printing technologies, specifically lithography printing, inkjet printing and EHD printing and their applications in biomedical and electronics devices, as illustrated in Figure 1 below. The precision printing technology has capabilities of altering the physical and chemical properties of materials by introducing micro/nano structures, which cannot be achieved through conventional fabrication methods. 

## 2. Micro-/Nano Printing Technologies

### 2.1. Stereolithography Printing

Stereolithography (SL) is a layer-by-layer 3D printing technique that forms objects by curing liquid materials by photoirradiation [19], as shown in Figure 2a. SL is an innovative technique that uses a lens to focus the laser beam and achieve resolution in the micrometer range [20]. The light centered 3DP method has been utilized in the creation of biomedical microdevices such as tissue scaffolds because of the high resolution and precise construction capabilities, which is a result of controlling different aspects of the photons applied [21]. Popular stereolithography based printing methods on the micro-/nano scale comprise of two-photon stereolithography and dip pen nanolithography.

#### 2.1.1. Two Photon Polymerization (TPP)

Two-photon SL based on photopolymerization (TPP) can fabricate polymeric and less commonly metallic microstructures at a resolution of 100 nm [23,24,25]. The working principle of TPP involves a laser beam concentrated onto a small volume of a photopolymer by utilizing high numerical aperture objectives. This solidifying mechanism enables the fabrication of freeform features such as helices. TPP technology has many advantages in producing highly precise 3D structures over conventional small scale methods which include high spatial resolution, easy handling, repeatability and variety of material choice [26]. However there are challenges associated with the TPP technique when applications are biomedical based. Hydrogels are commonly used in cell and tissue engineering but when 3D scaffold hydrogels are produced by TPP they have a low spatial resolution because of a high water content. This makes precise manipulation of topography and porous features at the nanoscale difficult to achieve [26]. Two step absorption is preferred to the previous technique of two photon absorption in the TPP printing process as the latter mechanism although is capable of nanofabrication [27], it is associated with issues around cost, reliability and higher order processes. The absorption method possesses that same quadratic optical nonlinearity while offering miniaturization and cost reduction of 3D nano scale printers [28]. There are also limitations to suitable materials due to the need for resins with a low viscosity, as well as the expensive cost of the equipment. A challenge associated with photon stereolithography is the low throughput of the process, which can be a serious problem when mass production is required [29]. 

Applications of this technique have been reported to cover micro and nano photonics, micro-optical elements and scaffolds for tissue engineering due to its high resolving powder and flexibility [30]. Thompson et al. studied the effect of various control parameters on the minimum laser power needed to achieve polymerization of a 3D polycaprolactone (PCL) scaffold to successfully support retinal cell replacement. TPP was the technique of choice as it is suitable in achieving precise microstructural control of PCL material in a biocompatible platform [31]. Stoneman et al. used TPP as an improved method to produce microfluidic device masters designed to trap individual yeast cells. TPP is advantageous to the trapping region of the device due to its intricacy in designing a 20 µm channel [32].

#### 2.1.2. Dip-Pen Nanolithography (DPN)

DPN operates by utilizing capillary forces to carry ink on a sharp tip to a substrate [6]. by employing CAD to control the tip movement to create complex structures [33]. The technique, as seen in Figure 2b, is based on atomic force microscopy (AFM), as the AFM tip delivers the ink. The size of the tip is attributable to the micro-/nano scale printing technology, as it can range from tens of nanometers to several micrometers [34]. The inks used can consist of small molecules, polymers, DNA, proteins, nanoparticles and peptides, which are directly transferred to the substrate with accuracy to a sub-50 nm scale [33]. Micro-/nano structures are introduced via patterns which can achieve a resultant resolution of 15 nm [35]. The generated patterns may comprise of organic molecules and catalysts, which can be used to control the assembly of nanomaterials such as gold nanoparticles and carbon nanotubes. Although DPN offers many advantages over conventional 3DP especially due to its high resolution, material compatibility, cheap cost and capability to operate in ambient conditions some challenges have been documented as low throughput and poor reproducibility [33]. Additional advantages and disadvantages of DPN technology are listed in Table 1 below.

DPN is effective in applications such as molecular electronics, material assembly and biological recognition [39]. Li et al. utilized DPN technology to design versatile sensing patterns with multiple compositions and structures for biosensing. The sensors proved to have high sensitivity, selectivity and fast response in detection and cell recognition. The study adapted the micro-/nano scale into the design by successfully depositing a 30 nm molecule based line on a gold thin film [40]. Meanwhile Liu et al. developed 3D DPN via rapidly UV-curable liquid co-polymer ink with appropriate viscoelastic properties. The printing structures employed were patterning dot and line pixels which resulted in micrometer high 3D structures. Results gathered consisted of printed lines with a height of 12 nm and width of 376 nm, in comparison the patterned dots generated measured to have a base size of approximately 35 µm [41]. Hence the process proves to be successful in producing structures on the micro and nano scale.

### 2.2. Inkjet Printing

Inkjet printing is a highly accomplished process in manufacturing with applications covering solders for electronic devices and thermoplastics for free-form fabrication [42]. The procedure operates by printing a liquid binder onto thin layers of powders on shapes determined by the design software [43]. There are two types of printer heads that can be used for micro/nano scale printing which employ the two techniques of inkjet printing known as piezoelectric and thermal printing, as illustrated in Figure 2c. Both printers are known for their precise sedimentation of liquid with accuracy and speed [44]. Inkjet printing processes have the ability to print layers as thin as 2 µm [45]. by controlling the critical parameters identified as the applied voltage, working distance and collector velocity [46]. Microfabrication by inkjet printing has been often require expensive equipment and a supporting or protecting layer. However, Zheng et al. proposed an economical method based on ice printing which can form structures with a maximum height of 2000 µm. The innovative method does not require additional support or removing processes and the need to introduce additional chemicals. The ice printing inkjet technique is promising as it has great potential in producing porous scaffolds of salt metal nanoparticles. [47]. The geometric ability on inkjet printing can be controlled by moving the substrate during the printing process, Jung et al. demonstrated this by achieving nanopillars with a feature size of 85 nm. A correlated challenge to inkjet printing lies with the limitation in geometries that can be achieved using high purity metals. However, Jung et al. successfully conducted experiments that produced 3D arrays of metal nanostructures with flexible geometry with nanoscale features as small as a few hundred nanometers [48]. The decision on which method of inkjet to use is dependent on the desired properties of the final part [43].

#### 2.2.1. Piezoelectric Inkjet Printing

Piezoelectric inkjet printing is based on the deformation of piezoelectric transducers with an applied voltage resulting in an induced mechanical vibration, as illustrated in Figure 2c. This mechanical vibration is enough to break the liquid surface tension to produce droplet ejection. Piezoelectric inkjet printing is desirable because of its capability to generate and control uniform droplet size and direction of ejected liquid meanwhile avoiding subjecting the binder to heat stressors, which is associated with thermal inkjet printing [43]. Piezoelectric inkjet printing is utilized to overcome issues associated with 2D ink writing, which is limited to a larger scale ability. The process is an appropriate method in manufacturing micro and nano structures, often by including nanoparticles within the printing ink for micro-reactors, micro-wires, micro-electrodes and nano-capacitors [49]. As piezoelectric is a bottom up approach, it possesses high aspect ratios within the micrometer scale meanwhile using minimum material possible. The degree of freedom enables designs to be printed in all three dimensions, proving it to be advantageous over conventional methods (e.g. MEMS) in addition to the inexpensive costs of printing parts. The ease of control of the printing parameters, e.g., flying trajectory and solidification degree of the jet [46] allows the ejection of material to be manipulated, this is an approach in developing micro/nano structures onto a substrate. A limiting factor to piezoelectric printing is the issue associated with horizontal free standing structures over large gaps, therefore printing over large flat gaps is ideally produced via continuous filament writing. The major challenge linked to this style of inkjet printing is clogging of material within the 50 µm [49]. orifice opening which is often a result of drying during downtime [50]. This produces unstable or non-existent droplet ejection which often occurs with high substrate temperatures and high solid inks. Due to the controllability of design and the precision of this technique, it is applicable within the field of microelectronics. For example, Kullmann et al. utilized piezoelectric inkjet printing of gold nanofluids to create 3D microstructures. The technique was deemed as a suitable approach due to the high throughput and large scale bulk production capability [49]. Kuznetsova et al. used piezoelectric plates within the printing head to utilize piezoelectric printing with silver nanoparticle ink due to its excellent resistance to oxidation, conductivity and low cost. The resultant device was a 4 µm thick electrode as a component for plate acoustic wave devices [51]. The small scale device is supportive evidence of the structural capabilities of piezoelectric printing.

#### 2.2.2. Thermal Inkjet Printing

The thermal inkjet printing technique illustrated in Figure 2c operates by a pulse of current passing through the heating element at approximately 300 °C for a few microseconds to vaporize the ink [52]. This causes a pressure increase and forces an ink droplet to deposit onto the substrate. Thermal printing is advantageous because the solid content in the ink can be high (20 wt%), therefore increasing the print efficiency and significantly lowering the printing cost [53]. Thermal inkjet printers can deposit ink volumes between 10 and 150 pL from the nozzle, however this is dependent on temperature gradient, current pulse and ink viscosity [54]. The limiting factor to thermal inkjet technology is the large droplet size which is typically as large as the printing nozzle itself and the need for the printing ink to withstand high temperatures [55]. Therefore, inkjet printing has been advanced to electrohydrodynamic printing as this technique has the ability to print at a dimension smaller than the nozzle at room temperature and high speed [6]. Additionally, the precision of the method is typically limited to the tip diameter, forcing it to be an unsuitable procedure for some applications.

Thermal inkjet printing has a strong application within the biomedical field, particularly within thermal printing of organs and tissues, as Cui et al. demonstrated by printing at 18 μm thick. The biocompatibility is of major importance, hence why thermal printing is more suitable as cells are only subjected to heat for 2 µs resulting in a cell viability of 90% [54]. However other applications do exist such as electronics. Setti et al. successfully printed a thin film measuring 230 nm via thermal inkjet printing with a novel application towards bioelectronic circuits [56]. Huang et al. employed thermal over piezoelectric printing because of its lower cost, higher nozzle density and printing speed. The resulting product used binder and graphene oxide ink with microstructure properties of line widths ranging from 192–262 µm and surface roughness ranging from 4 to 7.6 µm, which acted as a substrate for LED circuitry [57]. Organ printing and tissue engineering is achieved via thermal printing, as demonstrated by Cui et al. This study positively printed microvascular fibers with a diameter of 93 µm, which proliferated along a fibrin scaffold. By simultaneously depositing living cells, nutrients, growth factors and drugs within biomedical scaffolds at the right time and location, creates promising possibilities within biomedical engineering [58].

### 2.3. EHD Printing

Electrohydrodynamic (EHD) printing is controlled by computer procedures utilizing electrostatic forces to prepare fibers or particles with sizes from nano-range to a few microns through electrically charged printable ink [59], as il-lustrated in Figure 2d. EHD printing technology provides an effective and low-cost approach for the fabrication of micro-/nano structures [60,61,62], while allowing the application of biomedicine [63], quantum dot LEDs [64], flexible/stretchable electronics [65,66], microbial and sensor chips [67]. This technology has a wide range of materials applications and higher resolution than inkjet printing [68]. The mini inner size of nozzle is better for high resolution, which ranges from ~100 nm to several μm [68,69]. The high resolution can be achieved by exploiting electric force between a miniature nozzle and substrate to generate the droplets as well [68,70]. Typically, EHD printing system contains syringe pump, injector, metal nozzle, computer control stage, high voltage, substrate receiver, and printable ink (Figure 3a). A back-pressure supply (e.g., injection pump) extrudes the ink to the tip of a nozzle, then the electric field between the nozzle and substrate leads to the accumulation of mobile ions to gather at the ink surface of the meniscus (Figure 3b) [69,71]. The Coulomb force between these ions deforms the meniscus at the nozzle end into a “Taylor cone” (conical shape) at the liquid–air interface. Taylor cone sharpens upon the increase of the electric field force. Once the electromagnetic (Maxwell) stresses (external press force *Fp*, gravity force *Fg*, the electric field force *Fe*) exceed the surface tension force *Fst* and viscous force *Fv* of the ink, droplets are ejected from the cone [72]. (Figure 3b) Previous research has found various jetting behaviors controlled by different experimental conditions [63,67,71,73,74]. A numerical model based on a systematic study of electrified jet printing was stablished via a Volume of Fluid (VOF) method by Rahmat and co-workers. They found the printing process is mainly controlled by the external voltage between the nozzle and the substrate, and the inlet feed rate [75]. Hartman and co-workers based a physical model to calculate the shape of the liquid cone and jet based on the gravity, viscosity, surface tension, electrostatic force, back stress of cone [76]. Depending on solution property (viscosity, conductivity), electrical, mechanical (back pressure, flow rate), and surface tension forces, the EHD jetting modes are summarized ten types according to different geometries [67]. Collins et al. summarized the EHD jetting modes included dripping mode, pulsating mode, stable cone injection mode, and complex injection modes (e.g., oblique injection, double injection, and multiple injection) [77,78]. Several common modes are shown in Figure 3c. Among them, dripping and jet modes are the most typical, other modes derive from dripping and jet modes [79]. A dripping mode is observed using relatively small voltage with a low flow rate of the ink, while a jet mode is formed with a high flow rate. Even a dripping mode can be controlled without applied voltage and with low viscous ink. A slight increase in voltage based on dripping mode, a micro dripping mode can be obtained with a much smaller size than the nozzle. Unlike the dripping mode, the pulsating jet mode does not shrink the meniscus after the droplets are separated [80]. A pulsating mode is formed by a slight increase in voltage or flow rate. In this mode, streams of obvious droplets are extruded by repeated formation and disappeared of the Taylor cone. Jangung Park et al. captured time-lapse images of the pulsating liquid meniscus in one cycle. When the voltage is lower than the critical voltage or the flow rate is lower than the minimum flow rate requested for a cone-jet mode, a pulsating jetting behavior can be generated [81]. With the slight increase of flow rate, a spindle-shaped fragment can be generated before a jet. A spindle mode has a feature of an extremely high electric field more than a certain critical value, in which the flow rate approximately equals to jetting rate. With a further increase of applied voltage and viscosity of ink, a continuous and stable Taylor cone presents with a continuous jet ejected from the nozzle at the apex of the cone (stable cone-jet). With an excessive increase in voltage, complex jetting behaviors occur (titled, multiple jets, etc.). Most of the other jetting modes are uncontrollable except for a stable jet mode or pulsating printing mode. Therefore, electrospray, dripping (micro dripping) and EHD direct writing are the most helpful for micro/nanoscale structures and devices because of the precisely controlled of droplets or filaments. Back in 2008, EHD-jet-printed electrodes were used for source and drain fabrication on flexible plastic substrates to prepare SWNT based flexible thin-film transistors (TFTs). The printed feature sizes were high in a range from ~240 nm to ~5 μm [82]. Wang et al. utilized EHD direct writing to fabricate a stretchable piezoresistive sensor with a hierarchical porosity and multimodulus architecture for health monitoring [66]. Kim successfully used an electrospray system to fabricate biodegradable, multi-shell capsules for use as a drug delivery system [83].

#### 2.3.1. Controlling Parameters

To accomplish stable printings and obtain ideal micro-nano structures, several controlling parameters play key roles and need to be set reasonably, such as process parameters (flow rate, applied voltage, collection distance, moving speed of X–Y stage), ink physical properties (surface tension, conductivity, viscosity, concentration), and nozzle structure (nozzle diameter and design) [84,85,86]. Recent report indicated that among all parameters, the voltage, the permittivity, the nozzle height above the substrate (collection distance), the viscosity, and the conductivity are the most effective parameters on the process outputs [87,88,89].

#### 2.3.2. Process Parameters

Different types of voltage lead to various EHD jet modes and a size change of droplets [78]. Pan proposed a numerical simulation model for the whole process of droplet generation of E-jet printing and verified the accuracy with experiments. This model accurately predicted the effect of applied voltage on jet state, shown in Figure 4a [85]. Su et al. showed the diameter of printed droplets decreased with the increase of applied pulse amplitudes, while increased with the increase of applied constant DC voltage [90]. Lee captured the jet state under different amplitude of pulsed voltage [91]. Furthermore, the change of voltage value may enlarge or diminish the diameter of outputs based on different ink formulas. Zhang et al. found the size of obtained outputs decreased with the increase of applied high voltage when fabricating graphene oxide for energy storage and sensing via EHD printing [92]. Moghadam et al. indicated the obtained beads diameter slightly increased with the increase of applied voltage beyond a certain value [93]. Hence, the applied voltage needs to be matched with other optimized parameters to generate a specific micro-nano structure. 

The flow rate has effects on the structure and size of generated fiber or particles. A suitable flow rate contributes to sufficient volatilization, while superfluous flow rate leads to bad structures such as beaded fibers. The flow rate of EHD printing can be calculated by, where Q is flow rate, ΔP is the pressure drop, μ is the viscosity of the ink, dN and L are the diameter and length of the nozzle, ε0 is the permittivity of the free space, γ is the surface tension of the air-ink interface, and E is the strength of the electric field [73]. From the formula, flow rate strongly depends on several other parameters and they have synergy on droplets and structures. Previous work by Lee et al. also matched with this conclusion [94]. They proposed an approach to obtain the Taylor cone of the cone-jet mode and evaluate the jetting stability. In this work, when ink viscosity, density, permittivity surface tension, conductivity et al. are respectively certain values, EHDP states change with the variations of applied voltage and flow rate, shown in Figure 4b. If other parameters change, the experimental results will also vary. Kim et al found when the flow rate is increased, both the necessary potential and the pattern width increase for both tip cases with a tip inside the nozzle and without a tip inside the nozzle [97]. 

Another important process parameter is the moving speed of the X–Y stage or needle. Liashenko summarized the printing speed (µm^3^ s^−1^) as a function of the feature size (voxel size, µm) for several 3DP techniques (Figure 4c) with submicron resolution and found each order of magnitude increase in printing resolution results in 4 orders of magnitude slower printing [95]. They proposed the electrostatic jet deflection strategy to improve the printing speed and resolution, being capable of printing feature sizes down to 100 nm with unprecedented printing speeds up to 10^5^ µm^3^ s^−1^. In EHD printing, when the jetting speed is over the moving speed of the X–Y stage (or needle), the skewed cone jet can be obtained. If the jetting speed matches the moving speed of X–Y stage (or needle), a straight cone jet is generated. When the jetting speed is higher than the moving speed of the X–Y stage (or needle), wavy filaments are produced [80]. Previous studies found the line width of EHD filament decreased with the increase of the moving speed of the X–Y stage (or needle) [92,98], while He et al. gave the opposite result [99]. That is because they used different solutions and parameters to conduct experiments.

The distance between needle and substrate also plays a vital role in the printed structure and size of EHD outputs. For example, a working distance of a too low value leads to incomplete volatilization or particles aggregation. On the contrary, a working distance of a too high value needs a sufficiently high electric field and attributes to particles or fibers with a large size. Figure 4d shows the domains of EHD jet modes at different flow rate Q, working distance Z, and applied voltage V. The dashed areas are EHD direct-writing domains in which the stable segment length (hs) was smaller than Z [96]. Researchers can adjust Q, Z, and V to obtain specific targets based on this result. However, several parameters have thresholds to EHDP. For example, when the applied voltage reaches a threshold, electrical breakdown may even occur. Besides the process parameters above, Bu et al. studied the effect of substrate on continuous EHDP and verified that the conductive and semi-conductive substrate is the good substrate for collecting the patterns from EHD printing [100]. Because the process parameters are interplayed with each other, it is necessary to predict experimental results according to a simulation model to save time and improve efficiency. For instance, Collins used a Lagrangian model to simulate the electro spraying of nanoparticles on a surface and studied the effect of the spatial distribution and density of surface charges [69]. Mohammadi studied the formation of a droplet in an EHD printer under a pulsed electrical field using a new numerical model; the experimental results agreed well with the predicted results [87].

#### 2.3.3. Ink Physical Properties 

Physical properties of the liquid such as surface tension, viscosity, density, and electrical conductivity play a key role in the formation of final structures under the electrical assisted atomization process [78,101]. Since the resulting products are also determined by the properties of the ink. The ink should have a suitable concentration and be stable without aggregation [102,103]. When the ink concentration increase is too high, the droplet volatilizes too fast to form an ideal micro-nano structure. When the concentration exceeds the critical printable value, the ink is too viscous to be extruded due to the formation of large aggregates. Yuk et al. explored the effect of the PEDOT: PSS ink concentration from 1–10 wt% on printing and found that the intermediate range of PEDOT: PSS nanofibril concentrations (5–7 wt%) provided optimal rheological properties and 3D printability (Figure 5a). With the increasing concentration of the PEDOT:PSS nanofibrils, the suspensions gradually transited from liquids to thixotropic 3D printable inks (Figure 5b) [104]. The concentration can be adjusted by changing the ratio of additive particles or main substances (polymer, metal et al.). For instance, by changing the concentration of AgNWs in the precursor IZO ink, a one-step gravure-printed IZO/AgNW electrode showed a sheet resistance of 9.3 Ω sq^−1^ and optical transmittance of 91% [102].

The viscosity can also affect the resolution and the quality of the printing. Yu et al. found that viscosity increased the stability of Taylor cone jet [105]. However, too high viscosity results in jet diameter increasing and further induces thick jet, which is not suitable for producing high-resolution patterns. If the ink viscosity is too low, maybe no fibers or particles are generated. The viscosity is based on the ink concentration to a certain extent and thus can be changed by concentration variation. Moghadam showed the solution viscosity increased with the increase of alginate concentration [93]. Kwon et al. reported that the droplet diameter decreased when the polymers had high molecular weight [81].

Surface tension also plays a key role in the formation of droplets. Surface tension is the main parameter determining the time of droplet generation, as liquid breakup is a result of electrical forces overcoming the surface tension of ink. Notz et al. reported that the mode of breakup of electrified conducting drops changed from dripping to micro dripping as the relative ratio of electric force to surface tension force increased [97]. Weber showed that the optimum wavelength for the jet breakup depended directly on the viscosity and inversely on the density and surface tension of the liquid [93]. Kwon et al. showed that the jetting pulse duration increased with increasing viscosity and decreased with increasing conductivity and surface tension [80]. On the other hand, the surface tension and viscosity have a direct relation and the electrical conductivity and density have an inverse relation with the size of beads. The surface tension of the various solvents critically determined the properties of the cone-jet mode [106]. Ink with a surface tension over 50 mN m^−1^ would not generate a stable jet, because the electrical field would be exceeded when voltage is applied [84]. Thus, the common ink solvent is ethyl alcohol (EtOH), acetone, methyl alcohol, acetic acid (CA), dimethylformamide (DMF), dimethyl sulfoxide (DMSO), etc. In addition, the surface tension can be adjusted by using additives in ink solution, as well as viscosity and electrical conductivity [69]. In the ink of poly(ethylene oxide)/silver nanowires, the incorporation of poly(ethylene oxide) (PEO) increased the viscosity, reduced the surface tension of the ink, aided jet formation [69]. 

Bae et al. demonstrated that conductivity significantly affects structures of the meniscus [106]. EHD ink with higher conductivity conduces to minishing the droplet diameter and increasing the droplet frequency. Higher conductivity allows more effective fluid charging in the same electrostatic field, resulting in a faster transition toward the conical phase, by promoting the charge convection flow; in turn, this yields smaller droplets with higher frequency. Liquids with conductivity lower than 10^−10^ S m^−1^ would not be able to carry sufficient charge, leading to an unstable jet [84]. However, EHD ink with too high conductivity and viscoelasticity leads to whip motion and is not suitable for stable EHD printing.

#### 2.3.4. Nozzle Structures

The design and structure of the nozzle also affect the resolution and morphology of the resultant droplets (Figure 6a) [107,108]. A jet pulled from nozzle orifice by electrical field force jet has much a smaller size than the nozzle diameter [85,109,110,111]. Yu et al. obtained jets sizes of about 10 µm using a nozzle over 100 µm in diameter [112]. Park et al. printed DNA dots with diameters less than 100 nm using high-resolution EHDP technique with a nozzle of 500 nm in diameter [113]. Actually, it is possible to shrink down the nozzle size and printing resolution further by designing newer structures and setting matched process parameters. This will be a greater challenge, but it could contribute to realizing more effective E-jet printing [109]. Previous modifications on the nozzle design further improved EHDP effects. Kim et al. improved the resolution of EHD printing by applying a hypodermic needle instead of a flat needle and a line of 0.7 μm width was made [114]. The hypodermic needle generates a smaller meniscus thus the line width decreases nearly 95% compared to the line that was printed by the flat needle. In addition, UV-treated needle surface helps the ink slide down the wall more effectively [114]. A tip-assisted EHD printing method was proposed by Zou to improve the resolution nearly five times, shown in Figure 6b [115]. Figure 6c shows that the EHD printing system consists of the added metal ring [116]. The added metal ring can effectively eliminate the standoff limitation for the printing of 3D structures [116,117,118]. To overcome the low production rate, multiple nozzles can be used to enhance efficiency [70,109,119,120,121]. In multinozzle EHD printing, it is critical to control the uniformity of individual jets, jet sizes, and positions of each nozzle [122]. Recently, new designed nozzles for printed structures (e.g., core-shell, Janus, and multi-core structures) consisting of micro-scaled, multi-layered and multi-materials with simultaneous forming of multicompartments in single printed fiber have been achieved, as shown in Figure 6d–f. 

## 3. Printing Materials

### 3.1. Metals

The printing of metal is heavily popular within the production of electrical components; this is due to the incorporation of nanoparticles. Gold and silver are examples of nanoparticles used in applications such as electronics and photonics due to their high conductivity, oxidation resistance and chemical stability [6]. Li et al. successfully conducted electric-field driven micro scale 3D printing using nano-silver paste for electrode applications [125]. Metals are advancing in the biomedical field due to the mechanical demands of hard tissue, therefore by introducing metal, the demand for support and immediate mobilization of patients can be met. Clinical implants including but not limited to one of the following, metals, stainless steel, cobalt-chromium alloys, titanium alloys or nitinol make up 70–80% of existing clinical implants [126]. It is difficult to manufacture 3D printed metal structures that are biocompatible, due to the mismatch of properties between metals and bone which can result in stress shielding, bone resorption and inevitably, failure [126]. However, their excellent strength is too promising to completely eliminate the option of 3D printing metals. Hence, Nocheseda et al. extrusion printed metal powder mixed with biodegradable hydrogel to form the printing ink, containing 75–80% metal which produced a product consisting of 100% metal after sintering of the hydrogel [127]. The choice of metal used for 3D printing is dependent on a number of factors, of which include the application which therefore considers the material properties, i.e., conductivity, melting point, mechanical properties and magnetic behavior. The metal of choice also needs to be compatible with the 3D printer, as not all printers can successfully print metal, especially far from micro/nano printing using metal materials, this is usually due to the high melting point of some metals which the printers are not equipped to attain. Therefore, metals used as additives to form composites for 3D printing have involved polymers filled with metal nanoparticles to formulate nanocomposite filaments with the intent to improve features such as thermal resistance and mechanical properties [128]. The reinforcement of micro/nano filaments has a particular role within the aerospace and automotive industry. The customizable macrostructure and adjustable mechanical properties are favorable characteristics in manufacturing large scale composites with advanced geometries and mechanical tolerances.

Among the conductive materials for electronics fabrication, metals represent the main choice. Pure metallic powder such as silver (Ag) and titanium (Ti) are expensive and usually as additives to enhance the conductivity of polymeric systems to form composites, this part is reviewed in 3.3. Liquid metals have been used for stretchable wires and interconnect, soft sensors, self-healing circuits, and conformal electrodes [129,130]. Thereinto, gallium and its alloys with low toxicity are commonly used for electronics. Using inkjet printing, the gallium-indium (EGaIn) nanoparticles (NPs) suspension was printed onto an elastomer glove surface to form arrays of strain gauges with intricate wiring and contact pads by Boley et al. [131], shown in Figure 7a. Low-melting-point metals can be also used to fabricate electronics. The previous research [65] developed three molten metal inks (Field’s metal, Wood’s metal, and solder) to achieve metallic conductors fabrication with sub 50 µm resolution using EHDP, shown in Figure 7b. In their work, high-resolution EHDP and molten metal make a high-density touch sensor array possible. Metallic NPs and nanowires (NWs) have also been explored and used as major materials for printed electronics. Al, Ag, Au, and Cu nanowires are especially considered to be the best candidates for printable electronics because they can form films at low temperatures and in good conductivity [102]. AgNWs [132] and NPs [133] have the highest conductivity at room temperature, are the most widely studied metal nanomaterials as conductors or electrodes. Park et al. utilized Ag NPs to design electric circuits directly on the electrospun fiber mat by nozzle printing, inkjet printing, and spray printing, fabricating a highly stretchable antenna, a strain sensor, and a highly stretchable light-emitting diode [134]. Cui et al. also applied Ag NWs on fabricating heaters and electrocardiogram (ECG) electrodes and demonstrated the potential of high-resolution EHDP technique for AgNW-based flexible and stretchable devices [98]. Although conductivity (also very high) of Cu is 6% less than that of Ag, it is much cheaper. Thus, CuNPs and CuNWs for printed electronics are also attractive. Additionally, 3D metal structures based on metal NPs and metal NW networks with large length-to-diameter aspect ratios show high transmittance, good conductivity, and excellent mechanical compliancy [102]. Many metal-oxide semiconductors and the corresponding NPs/NWs have been also used for sensors, especially for gas sensors. Among them, transition-metal oxides such as ZnO, TiO_2_, V_2_O_5_, and WO_3_ are used as piezoelectric sensors, photoanodes, catalysts, and chemical detectors [135]. Non-transition-metal oxides, such as Al_2_O_3_, SnO_2_, In_2_O_3_, and Ga_2_O_3_ and ITO, which show good conductivity and high sensitivity, are useful for multifunction sensors [136]. To optimize the rheology of inks as well as improving their electrochemical, conductive and mechanical properties, adding nanomaterials as fillers has been proven to be an efficient way [137].

### 3.2. Polymers

Polymers, such as polycaprolactone and polylactic acid, are suitable materials for 3D printing due to their low cost, elasticity, lightweight and flexibility. They are often a desired printing material for biomedical applications due to the wide range available with bioinert and biodegradable properties that have been FDA approved for biological applications. Extrusion based printing is best associated with the use of polymer, of which PLA (polylactic acid) is most popular due to its low cost, biocompatibility, processability and degradation rate [138]. Liu et al. printed a PLA platform for drug screening and toxicity evaluation due to its straightforward structure, low cost and convenient integration of 3D cell culture and activity evaluation [139]. The current limitation to polymeric biomaterials is the ability to mimic biomechanical functions, hence the introduction of ceramic material is often employed to create a composite material with advanced properties. Polymers are a favorable material in producing electronics due to the design flexibility. Soft sensors are advantageous over rigid sensors as they can bend, flex and absorb shock, which are key attributes in wearable technology [140]. Emon et al. 3D printed a curved surface pressure sensitive polymer membrane sensor, with carbon nanotube based conductive electrodes. A multi-material extrusion printer was used to print the sensor components successfully in a curved structure to fit its application to be placed on the fingertip [140].

A wide range of polymeric materials can be available in 3DP for electronics (including semiconductors, electrodes, dielectrics, sensors). Generally, pristine polymers are seldom used and usually doped with other inorganic materials to form functional composites, which is specifically expanded on in Section 3.3. Polymers can be dissolved in solvents or melted at a certain temperature to prepare printable inks that can be extruded out of the nozzle. Printable ink towards electronics is required to meet polymer properties such as conductivity, viscosity, surface tension. Moreover, different 3DP techniques have disparate demands of polymer properties. Unlike the quite stringent requirements of inkjet printing in printable inks, one specific advantage of EHD printing is its adaptability to wide materials, which supports the fabrication of high viscous inks [71]. Conductive polymers are inherently flexible and soluble, which potentially allow for compatibility with large-area solution-processing methods such as inject printing and EHDP [141]. Conductive polymers such as poly(3,4-ethyelenedioxythiophene): poly(styrene sulfonate) (PEDOT: PSS) with tunable conductive and thermal properties, polyaniline (PANI) and polypyrrole (PPy) with excellent conductivity are broadly utilized in printed flexible and wearable microdevices [142]. Among them, PDMS are suitable for EHD printed electronics. Recent studies focus on fabricating flexible pressure and strain sensor array [143,144], ion-gel-based transistor [145], flexible electrodes [146,147,148], organic thin-film transistor (OTFT) [149,150] etc. Although PEDOT: PSS is commonly used in printed electronics, the ink properties, geometrical pattern and array can be further optimized for targeted applications and multi-modal sensor integration.

Ionic conducting polymers, compared to insulating polymers such as polycaprolactone (PCL), polyvinylpyrrolidone (PVP), thermoplastic polyurethane (TPU), polymethyl methacrylate (PMMA), polystyrene (PS) and polydimethylsiloxane (PDMS), have a higher electrical conductivity thus are popular for sensor fabrication. Ionic conducting polymer polyethylene oxide (PEO) and other composite ionic inks are suitable for printing. While insulating polymers need to be added conductive nanoparticles or nanowires to satisfy the demands of electronic materials. However, a few special initial or doped insulating polymers can be directly used to fabricate electronics as flexible substrate or dielectrics, including PDMS, e-coflex, polyethylene terephthalate (PET), polyimide (PI). Song et al. used PDMS as flexible substrate and PEO as the main polymer-based sensing layer to prepare flexible pressure sensor via EHD technique, the sensor is respected to wearable and submersible sensor [151]. 

Polymer-based piezoelectric materials such as poly(vinylidene fluoride) (PVDF), copolymerized poly(vinylidene fluoride-trifluoroethylene) (PVDF-TrFE), and poly(l-lactic acid) (PLLA) have been used for flexible tactile sensors [152,153,154], especially for piezoelectric and triboelectric sensors.

In general, photopolymer materials are used for lithography printing and are divided into standard, structural, tough and durable, flexible and elastic, castable wax and ceramic, biocompatible, and bioink [155]. In the structural resin family, high temp resin is suitable for injection molding prototypes, heat-resistant fittings, hot gas, liquid piping, and electronics covers. Elastomeric polyurethane (EPU) has similar properties of low stiffness and extreme ductility to polyurethane (PU) elastomers, belonging to highly elastic and flexible polymers. The high elasticity makes PU/EPU be widely used in orthotics/aesthetics. While flexible polyurethane (FPU) is a semi-rigid material and suitable for parts withstanding deformation, bending, and compression. Other light-sensitive materials, such as elastomers with nano-silica, and conductive/dielectric elastomeric materials, are widely used for sensors, actuators and robots to improve human–machine interactions [156,157,158,159].

### 3.3. Ceramics/Composites 

Ceramics are conventionally designed by injection molding, gel casting or tape casting, and require the sintering of green parts at high temperatures to accomplish densification. These conventional methods result in high processing times and high costs. Creating micro-/nano level structures is difficult as molds are required in these techniques to produce complex geometries [160]. Machining of ceramics leaves them susceptible to defects such as cracking despite the hardness of the material, ceramics are brittle [160]. The 3D printing of ceramics alleviates the aforementioned challenges by improving design, reducing processing times and creating complex geometries with precise micro-/nano level structures. Ceramics are printed based on three variations of feedstock form, slurry-based, powder-based and bulk solid-based; some examples of ceramics used in additive manufacturing are SiO_2_, Al_2_O_3_, ZrO_2_ and SiC. Slurry based ceramics involve liquid or semi liquid systems with dispersed ceramic particles in the form of ink or paste, which is printed via photopolymerization, inkjet or extrusion [160]. Powder based ceramics are used in powder bed fusion 3D printing, whereby a loose powder bed of ceramic material is used as feedstock and bonded by liquid binder or thermal energy from a laser beam. Lastly, bulk solid-based ceramics are difficult to produce due to its brittleness [161]. The printing of ceramics is often in the form of a composite, i.e., a combination of ceramic and polymer/metal to improve properties that are lacking in ceramic, polymer or metal alone. For bulk solid ceramics to be printable they are loaded into thermoplastic binders and the process uses thermal energy to extrude the material layer by layer onto a platform. This is followed by binder removal and sintering to achieve densification [160]. The particle size of the printing ink in many 3D printing processes is a considerable factor in the accuracy of the resultant print. For example, in photopolymerization larger particle sizes reduces the rate of polymerization as a result of light scattering and absorption. Additionally in inkjet printing the particle size needs to be small enough to prevent clogging or blockages in the printing nozzle. However, if the particles are too small, agglomeration can occur. Therefore, when printing with ceramic based ink, it must be homogenously dispersed to pass through the printing head with ease. The solid content of ceramic inks for additive manufacturing can affect the materials rheological behavior [162]. It is worthwhile conducting rheological assessments of the material to determine the extrudability and shape fidelity [163]. Printing requirements that are a factor of rheological properties is the ability to retain the shape of the printing nozzle, be self-supporting and accurately reproduce the printing path. These conditions can be satisfied when the ink behaves as a fluid during printing and presents elastic behavior at rest [163]. Materials printed at micro/nano scale have unique properties in comparison to its bulk state, enabling the control and manipulation of properties on the atomic scale [164]. The printing ink is a key factor in achieving micro and nano scale features in a 3D printed design. By using a printing ink containing nanoparticles measuring around 10 nm, nanostructures can be achieved. Wen et al. produced nanostructures using silica nanoparticles to a resolution of 200 nm which demonstrated attractive optical properties [165]. Micro/nano printing has roles within healthcare, i.e., biosensors, therapeutics and implants, which maintain a supply to demand due to faster processing times. Additive manufacturing using printing inks that are suitable biomaterials has become an increasingly popular area within research and the development of biomedical devices. Bioglasses and Bioceramics are examples of biocompatible materials used as printing inks to form a patient specific design with tailored properties to serve their purpose in a biological environment. Bioglass and bioceramics offer advantages over Ca-P based materials for bone regeneration due to their high bioactivity, angiogenic properties and osteoinduction capabilities. These bioactive materials have the ability to induce antibacterial effects and cellular responses [166]. Micro and nano structures are necessary for biomedical applications, as they are required to truly mimic the microenvironment. Hence mesoporous active glasses has gained significant interest in science as proposed drug delivery systems due to the mesoporous structure offering control on release kinetics, and therefore offering a confined therapeutic treatment [167]. Printable inks such as carbon are easy to manipulate in terms of shape and size by controlling viscosity through operating parameters such as temperature, pressure and printing speed [164]. A summary of the classes of printing materials, alongside their advantages and limitations, can be seen in Table 2. 

Inorganic ceramics are inherently brittle and not suitable for flexible tactile sensors. The inorganic piezoelectric materials are commonly used for piezoelectric sensors, including the zirconate titanate (PZT) system, barium titanate (BaTiO_3_) system, and zinc oxide (ZnO) [153]. However, the extremely high melting point of many ceramics adds challenges to additive manufacturing compared with metals and polymers. Advances in materials science make it possible to photopolymer pre-ceramic polymers (PCPs) for transformation into polymer-derived ceramic (PDC) components via heat treatment. The pre-ceramic polymers can convert by pyrolysis into SiOC, SiC, SiCN, Si_3_N_4_, boron nitride, and aluminum nitride ceramics, typically at around 1000–1300 °C under an inert atmosphere [89]. Eckel et al. reported preceramic monomers with 3D complex shape and cellular architecture cured with ultraviolet light in a stereolithography 3D printer or through a patterned mask [168]. These polymer structures can be pyrolyzed to a ceramic with uniform shrinkage and virtually no porosity. These materials are of interest for propulsion components, thermal protection systems, porous burners, microelectromechanical systems, and electronic device packaging [168]. Additionally, ceramics nanoparticles can be suspended in a solvent, so that they can flow out of a nozzle [69]. This group of inks is mainly composed of silica (SiO_2_), alumina (Al_2_O_3_), zirconia (ZrO_2_), and hydroxyapatite (HAP) ceramic particles. A study reported a printed ceramic micropattern using polymer-based ceramic inks via EHD jet. The polymer-derived ceramics by near-field electrospray printing can become a useful addition to the toolbox of high-temperature MEMS [169]. Baldev et al. explored the polymerization kinetics of a series of suspensions filled with different ceramic particles (SiO_2_, Al_2_O_3_, ZrO_2_, and SiC), found that smaller ceramic particles were with better light-scattering properties in stereolithography. Additionally, ceramic powder can be added to a polymeric solution to form sensing materials for sensor. This method improves the complicated and time-consuming procedures of photopolymerization. For instance, Kim successfully presented the 3D printing of piezoelectric sensors using BaTiO_3_ (BTO) filled in a poly(vinylidene) fluoride (PVDF) matrix through electric in situ poling during the 3D printing process [170]. Potentially, the process will enable the low-cost mass production of composite piezoelectric devices for use in the sensor industry. Besides BTO/PVDF, other 3DP additives/PVDF composites are summarized in detail in previous work [154]. 

Compared to ceramic composites, polymer/NPs (NWs) composites are more popular for the fabrication of electronics due to their simple and economic advantages. Polymers doped with ceramic NPs (SiO_2_, BaTiO_3_), carbon material (carbon black CB, carbon nanotubes CNTs, graphene G), metallic NPs/NWs (Au, Ag, ZnO, Pt NPs/NWs) etc. are endowed higher dielectric constant, electrical conductivity, and sensitivity, thus are widely used in wearable, human–machine interaction, and healthcare fields [135,171]. Wang et al. added Ag particles into TPU and explored the threshold of Ag in TPU/Ag to fabricate TPU/Ag stretchable circuit for the high-resolution 3D Printing of stretchable piezoresistive sensors [66]. Huang et al. fabricated a graphene-based elastomer (PDMS/G) for sensors with tunable and high sensitivity was fabricated by 3DP [172]. Park et al. printed complex 3D structures with multiple functional inks (Ag, Cu, Co, and anthracene) used inkjet printing, shown in Figure 7c [173]. This work offers a promising strategy for highly integrated devices and indicates substantial promise for use in next-generation electronics [173]. The composites with their properties and 3DP techniques are summarized by previous works [135,153].

**Figure 7 micromachines-13-00642-f007:**
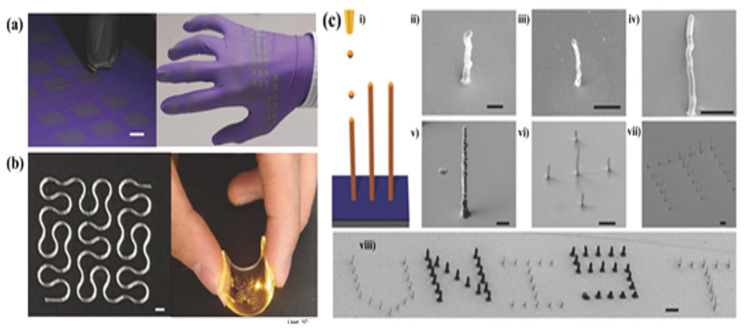
(**a**) Photograph of inkjet printing of EGaInNPs, scale bar: 5 mm [131]. (**b**) EHD printed circuit pattern on glass slide (left), scale bar: 500 µm; Healed conductor remained conductive when bended by hand (right) [65]. (**c**) SEM images of 3D pillar structures [173]. (**i**) Schematic illustration of 3D pillar printing. (**ii**) Printed Ag pillar. Scale bar, 2 μm. (**iii**) Printed Cu pillar. Scale bar, 5 μm. (**iv**) Printed Co pillar. Scale bar, 5 μm. (**v**) Printed anthracene pillar. Scale bar, 5 μm. (**vi**) Cross-shaped array of five anthracene pillars. Scale bar, 10 μm. (**vii**) Letter “E”-shaped array of anthracene pillars. Scale bar, 10 μm. (**viii**) “UNIST”-shaped array of Cu and anthracene pillars. Black and white pillars in the array are composed of anthracene and Cu, respectively. Scale bar, 10 μm.

**Table 2 micromachines-13-00642-t002:** Advantages and disadvantages of classes of printing materials and their relevant applications.

Printable Material	Example	Application	Advantages	Disadvantages
**Polymer**	PCL PLA PVP Hydrogel	Tissue engineering Electronics Medicine	Low cost Ease of use Biocompatible Good mechanical properties [174]	High MP makes it difficult to extrude Change in properties Shrinkage
**Metal**	Au Ag Ti	Microelectronic Biomedical Dental	Good mechanical properties Large variety of materials	Requires expensive printing equipment [175]
**Ceramic/** **Composite**	CaP Silica SiC	Microfluidics Dental Tissue engineering Pharmaceutics Aerospace	Low cost High resolution Microscale precise	Requires post processing [160] High cost Brittle

## 4. Applications

### 4.1. Biomedical

The 3DP on the micro/nano scale paves the future for customizable biomedical products that have patient specificity as accurate as the nanometer scale. Before the design capabilities of 3D printing, biomedical devices were produced in a ‘one size fits all’ approach, which is untrue, especially in meeting the requirements of the features of various genetic diseases. Therefore, by freeform fabrication of the products shown in Figure 8, such as drug delivery devices, scaffolds for tissue engineering and vascular structures, a biomedical device can be produced with higher functionality success rates. The recent progress of 3D printed drug delivery systems with micro and nano scale structures has enabled patient specific, controlled drug release to be possible. Meanwhile 3D printed scaffolds with micro and nano structures for tissue engineering promote biocompatibility and the growth of new tissue. Additionally, blood vessels have been 3D printed to manipulate vasculature structure to include porous features to allow the flow of cells and nutrients. 

#### 4.1.1. Drug Delivery System (DDS)

Three dimensional (3D) printing within the pharmaceutical sector is increasing in demand due to the possibility of developing patient specific drug delivery systems with controlled dosages and customized release profiles [177,178]. The cause for demand of personalized drug delivery systems is due to genetic variations in treatment responses, therefore by designing the drug/system the accompanied side effects of mass produced drugs can be alleviated to a lesser extent [18,179].

Microneedles designed by VAT polymerization [180] or stereolithography [181] are small devices comprised of needles that measure less than 1000 µm which facilitate transdermal delivery by penetrating the stratum corneum [182]. Uddin et al. fabricated microneedles using biocompatible polymers for cancer treatment via a combination of SLA and inkjet printing. The microneedle tips measured 100 µm with coatings from inkjet printing producing a consistent particle size of 100–110 µm. Nano-/micro fabricated microneedles have consumed the 3DP drug delivery sector. The novelty is attractive to both patients and healthcare professionals as the device can mitigate penetration pain and tissue damage as well as accurately controlling channels for administrating bioagents and collecting fluids [183]. There are some issues linked with the manufacturing of microneedles such as the uniformity of coating technology which is not always reproducible and accurate and may cause drug deposition on the microneedle substrate therefore resulting in waste and uncertainty regarding the exact drug dosage. Additionally, the limited drug loading makes the device unsuitable for the delivery of large drug amounts [184]. Similarly, Economidou et al. conducted a study which involved the SLA printing of microneedle patches for intradermal insulin delivery with an improved needle resolution of 25–140 µm and tip size 50 µm. The small tip size is optimal for pain free delivery. The needles also employed piezoelectric inkjet printing of 92 cycles to coat the needles with insulin which resulted in 350 µg per array [185]. The limitation of drug coating is suitable for small dose patches; however, the delivery system is not substantial in delivering large drug quantities.

Vivero-Lopez et al. identified a trend in issues arising from hearing aid wearers. The prolonged use of hearing aids can cause ear infections, therefore by loading the device with antibiotics the number of infections can be reduced. The study is based on VAT polymerization in fabricating patient specific medical devices with antibacterial properties. The 3D printed hearing aids consisted of a layer thickness of 25 µm, making the device small and lightweight which is an important factor for wearable devices. The antibacterial drugs were dissolved into the resin printing ink which produced a flexible device when printed. The high precision and resolution of the printing technique allows the design of the hearing aid to be manufactured precisely to the patients ear, making it much more comfortable than the standard hearing aid with limited sizes and shapes that are commonly mass produced [186].

In contrast to the latter study, coatings/drug loading can also be applied EHD printing. Site specific drug delivery is favorable over conventional drug delivery systems as toxicity and side effects can be reduced by manipulating the characteristics of particles via electrohydrodynamic atomization (EHDA), a relatively new technique for fabricating 3D printed structures. The method can control particle size, charge and morphology, which can influence how effectively drugs can enter tissue [187]. Ali et al. discussed how EHDA can produce structures at the micro and nano scale, with capabilities to possess high drug encapsulation and drug release kinetics based on structure driven control. EHD printing has been designated as a suitable technique in developing DDS with various sizes, morphologies and compositions within a single step processing method [187]. An EHD study conducted by Wu et al. involved EHD printing to manufacture a flexible multi-drug that can be easily folded into capsules. The flexible printed membrane contains foldable linkages which can unfold to increase surface area for drug release; this effect increases retention time which is required to reduce dosing. The resultant structure possessed PCL fibers measuring between 4–19 µm, with a mean of 10.5 ± 3.2 µm, therefore demonstrating the precision control of EHD printing [59]. In addition, EHD printing allows flexibility in material selection and has ease in modulating design parameters for micro/nano scale features. A broad active ingredient can be incorporated on demand at ambient temperature during the EHD printing. Multidrug can be encapsulated into printed structures in a single step and the structures have various geometries capable of control drug loading and release behavior to maximize the potential of drug co-delivery and the physicochemical properties of the resulting structures.

#### 4.1.2. Scaffolds for Tissue Regeneration

Extrusion, inkjet and photopolymerization are the best known printing technologies used to fabricate tissue engineering scaffolds [188]. The micro and nano scale structures that can be produced via various 3D printing technologies make it a favorable application for tissue generation. Creating a micro-/nano topography on scaffold structures is an important feature as it mimics the biological microenvironment which is encouraging for cell proliferation. 3D printing at a highly precise level allows surface structures to be modified such as roughness and hydrophilicity which play a large role in the success of cell adhesion and differentiation. Wang et al. utilized 3D printing of microscale surface roughness and also incorporated nanoscale features to enhance osteogenic differentiation. Results had shown that samples possessing a microscale surface roughness had much better hydrophilicity than smooth surfaces, therefore promoted cell adhesion [16]. The 3D printing of biocompatible micro structures is popular within orthopedic applications because the complex structures of bones can be modelled on design software, giving a more controlled micro structure that mimics the trabecular bone topography. X. Lie et al. recognized that the challenge lied with bone augmentation of bio-ceramics which is required for bone regeneration. Therefore, an extrusion based 3D bio-plotting technology was employed to fabricate a biomimetic hierarchical scaffold with interconnected porous scaffolds with a micro/nano hydroxyapatite surface. The printed ceramic fibers measured a diameter of 250 µm and the overall structure comprised of meso-pores (2–50 nm) and macro-pores (greater than 50 nm). The micro/nano surface favored adhesion and osteogenic differentiation [189].

Although the aforementioned printing processes are ideal for tissue engineering scaffolds, they are not optimal for high resolution porous structures. Stereolithography, digital light processing and TPP require photo-initiators which able higher process resolutions at higher precision. TPP is a highly favorable technique in fabricating precise scaffolds with biocompatibility for tissue engineering. The technique is advantageous in the micro realm due to its high resolution. Grasset et al. fabricated a spider-web like structure using TPP technology with microstructures via the movement of a laser beam by moving a piezoelectric platform. The structure width measured 1.2 µm at its widest and 1.1 µm at its narrowest. Additionally, the surface roughness was calculated to be 82.81 nm at its highest and 25.73 nm at its lowest. The overall results presented excellent cell viability of 66.3%, which was achievable through the surface roughness and microstructure design [188]. The fabrication of three-dimensional porous architectures are favorable in the design of biomedical devices as the modified surface is ideal for enhancing cell activity and infiltration in tissue repair and regeneration. [190,191,192]. Wang et al. successfully introduced a porous structure via VAT polymerization, extrusion and handheld bioprinting by using a micropore-forming bioink. Dextran was used as the porogen to generate a microporous structure measuring 43.7 ± 7.5 µm. The microstructure proved to be superior in cellular behavior in comparison to non-microporous structures [193]. Porous structures portray an interconnected network of material which can alter the transport pathways within a printed structure [194]. Aubert et al. reported novelty digital light processing of mesoporous multi- materials which avoids the calcification step that is required in the extrusion printing process of mesoporous structures. The methodology is additionally advantageous as the direct printing of mesopores enabled pore accessibility and control over the internal structure [194]. The challenge remains in producing sub-micrometer scale features as part of a larger polymer-based design, however combined digital light processing with polymerization induced phase separation to manufacture macroscopic polymer objects with a controlled inherent nano porosity is an alternative method for biomedical applications [195].

Biomimicry is an asset desired in biomedical engineering, particularly in a 3D manufactured scaffold for tissue engineering. It is understandable that a biomedical device constructed from a single material will not meet all the requirements to succeed in a biological environment. Therefore, by fabricating a biomedical device using multiple materials, the chemical and physical properties of the overall product will be an advancement upon the single material design. Multi-material design can influence factors of the microenvironment and is desired in establishing the functionality of engineering tissues [196]. Continuous composition gradients are desirable in scaffolds for tissue engineering which can be achieved via multiple material printing ink. This requires material dosing and mixing controls to modify surfaces to promote cell adhesion. Sinha et al. conducted this technique using a dual material print head to produce continuous composition gradients of thermoplastic scaffolds with improved mechanical properties and enhanced cell adhesion [197].

Orthopedics is the most common application in 3D printing tissue engineering as it typically involves a supportive structure with unique surface properties to promote cell adhesion and growth, and is primarily developed using bio-ceramics or hydrogels. The 3D printing of live organs and vasculature is much more complicated and relatively new. It is coupled with many challenges such as high resolution cell deposition, cell distribution and intervention with complex 3D tissues [198]. Obvious challenges include preventing cell damage during the printing process and maintaining the cell culture during slow printing processes. The demand for patient specific manufactured tissues is growing due to an expanding healthcare system and aging population.

#### 4.1.3. Vascular Structures

The fabrication of blood vessels is limited to the 3D printer’s capabilities as it must have the ability to print at a highly precise level to meet the dimensions of various blood vessels. Extrusion, inkjet, laser-assisted and VAT polymerization are all appropriate printing techniques that are capable of achieving structures as small as 10 µm, the smallest inner diameter across all blood vessels which is the capillaries [199]. The printing technique is dependent on the shape, size and in-vitro functionality of the blood vessel of desire. Bio-ink is used as the printing material which contains cells, nutrients and growth factors that are critical for determining resolution and biological properties in the 3D printing of blood vessels [199]. Hydrogel precursors are a common constituent of the bio-ink as it mimics the extracellular matrix, and therefore provides a suitable microenvironment for cell proliferation [199,200]. The fabricated vessels must portray appropriate mechanical properties, remodeling capability and anti-thrombogenicity to achieve successful integration with the surrounding biologically developed vessels [199]. Permeability and cytocompatibility are the biggest factors required in successful vascular supply systems. It is the inability to provide sufficiency of these factors is the biggest limitation in achieving functional blood vessel systems [201]. 3D printing of blood vessels offers many advantages in the medical field, as it is difficult to manufacture such small structures manually. Therefore, by utilizing ultra-precise methods, the resultant product can be produced to a scale that is highly reflective of biologically natural blood vessels. Challenges associated with 3D printing of blood vessels is commonly associated with suitable bio-inks and the delicacy of working with cells into a desired shape. 

Huber et al. successfully fabricated various 3D tubular structures, via SLA, to mimic blood vessels with a range of wall thicknesses. Three types of structures were fabricated, linear tubular, branched tubular and tubular structures with defined pores, all achieved via SLA. The linear structures had a wall thickness of 200 µm, meanwhile the branched structures had a wall thickness of 300 µm and the tubular structures exhibiting pores had a consistent pore diameter of 100 µm over the full tubular wall. A porous structure in the vessel walls is necessary to allow the diffusion of water through the tissue. However, the study noted that the pore size is not small enough to allow the passage of endothelial cells, and in that case it needs to be reduced to 20 µm. Therefore, it is apparent that the manufacture of 3D printed vessel systems is difficult as the level of precision must be achievable at a small micrometer level, or preferrable at the nanoscale level [202]. An existing challenge in relation to bioinks is the poor shelf availability as a result of fabrication and storage complications [202]. Additionally, the fabrication of 3D printed vascular scaffolds have insufficient mechanical strength and poor biodegradability as a result of the selected bioink [203]. Therefore, Xu et al. introduced a nano-enhanced composite bioink to produce linear and branched vascular scaffolds that do not express the properties of the latter challenge. The extrusion based, 3D structure design portrayed an interconnected microporous structure favorable for nutrient delivery and cell infiltration [203]. Interventional therapy is advantageous, as a consequence the demand has increased to treat vascular disease due to its high accuracy, minimal invasiveness and high efficiency [204].

### 4.2. Electronic

#### 4.2.1. Physical Sensor

These 3DP techniques have already been demonstrated as effective methods in physical sensors fabrication, with many sensor devices successfully fabricated. The 3DP physical sensors include tactile (touch sensors) [173,204], acoustic sensors [205,206], microcantilever sensors [207], transistors [145,148,149,150,208,209], strain sensors [143,171,172,210], pressure sensors [66,135,144,151,154,170,211,212] and (piezo resistance, piezoelectricity, capacitance, triboelectricity), photosensors [213,214,215,216]. etc. The printed touch sensor exhibits good flexibility and sensitivity [71]. The flexible ouch sensor is an excellent candidate for wearable, human–machine interface, and electronic skin. A capacitive acoustic transducer is an electromechanical acoustic system that usually consists of a fixed backplate electrode and a flexible diaphragm separated by an air gap to form a parallel plate capacitor [205]. An acoustic sensor can also be used for the navigation of drones, MEMS devices, strain sensors, and pressure sensors. Microcantilevers are micromechanical springboards that are anchored at one end and free at the other end, compatible with silicon electronics. These physical sensors can respond to changes in surface stress or adherent material and detect label-free various analytes [207]. Transistors, which can be fabricated by 3DP, are a vital part of electronics. A transistor includes OTFT, OFET, and an ion-gel transistor. A few applications of transistors are shown in Table 3 [208]. The 3DP organic transistors have been widely used as a switching unit-device in the logic circuit or display backplane, but also as a biological, photochemical, and electrochemical sensing device. Flexible (or stretchable) strain and pressure sensors are promising candidates for future generations of wearable intelligent electronics (touch sensing, health detection etc.). Interestingly, a kind of triboelectric sensor (triboelectric nanogenerator) can convert mechanical motions into electricity to supply energy for EHDP in turn [217]. 3DP endows these sensors with various functional materials and micro/nano structures. Thereby high sensitivity, linearity, fast response, and tunable pressure detection limit are available. The representative strain and pressure sensor by 3DP for wearable application are shown in Figure 9a [218]. Optoelectronic devices are a type of electronic device for light sensing, benefiting from high transparency, are broadly applied on missile launch detection, chemical/biological analysis, and optical communications [216]. McManus showed an all-inkjet-printed photosensor based on water/two-dimensional crystal formulations, which can be used for programmable logic memory devices, exhibiting the potential in biomedical applications [213]. Zou et al. printed graphene photodetectors through in situ EHD jetting. The photodetector had a response to the change in photocurrent of 0.22 μA, shown in Figure 9b [214]. Furthermore, Figure 9c shows EHD jetted devices for underwater sensing ability to detect various external aqueous pressures and real-time monitoring of water life in submersible systems and the applications of wearable soft electronics [151].

#### 4.2.2. Chemical Sensors

Chemical sensors are utilized to monitor chemical reactions via two main transduction mechanisms. The first one is the electrochemical method, including potentiometry, amperometry, and resistance. The second one is the optical method, which contains absorbance, colorimetry, fluorescence, luminescence, and reflectometry. Benefiting from the complex and customized micro/nano structures of 3DP, 3D-printed chemical sensors can be applied in sensing liquid concentration or components, gas concentration, pH variation, and measurement of biomolecules [219]. Based on the two main transduction mechanisms, chemical sensors usually include liquid sensors [220,221,222], gas sensors [223], PH sensors [224], electrochemical biosensors (detecting biological macromolecule), etc.

Here, we mainly introduce the 3DP application on electrochemical biosensors. The concentration or activity of a specific analyte in a sample matrix is detected via the coupling between biological macromolecule and electrode [225]. However, the detected environment of biological macromolecule is complicated, and the electrodes of electrochemical biosensor are of high requirements. 3DP with pretreatment can overcome these challenges with various materials and structures, resulting in enhanced selectivity of an electrochemical biosensor to specific molecules. A variety of biological macromolecules have been used within biosensors such as nucleic acids, antibodies and enzymes; the latter is the most widely used biological sensing element [226,227,228,229]. Adams et al. first presented an example of glucose dehydrogenase sensor fabricated by 3DP, and this preliminary study shows that the 3D-printed sensor platform holds promise for sensitive glucose detection [230]. Pumera et al. proposed 3D-printed stainless steel helical shaped electrodes with a gold (Au) film, which is used as novel electrode materials for the electrochemical detection of ascorbic acid (AA) and uric acid (UA) in aqueous solutions. The 3D-printed Au electrode generally performed better in terms of sensitivity and detection limits as compared to glassy carbon electrode (GCE) [231]. The electrochemical sensors used for antibodies or nucleic acids detection were also explored, examples are shown in Table 3 [229]. In summary, 3DP is an effective method of fabricating electrochemical sensors for the measurement of biomolecules [225]. In future, the detection of biomacromolecule based on 3DP should exclude the disturbed biomacromolecule. Furthermore, more flexible micro/nano structures of electrochemical sensors via simple 3DP should be developed.

#### 4.2.3. Hybrid Sensors 

Hybrid sensors integrate the functions of different functional sensors, which can simultaneously detect multiple signals. Hybrid sensors are the development tendency of 3DP sensors in the future. We summarize several hybrid sensors by different printing methods in Table 3. Recent studies use G as an active layer and an electrode in a pressure sensor [232]. Interestingly, these kinds of sensors have responses to temperature and pressure simultaneously in a single device. Furthermore, G can also serve as a semiconductor, enabling its application in transistor-based pressure sensors [233]. Deng et al. developed conductive self-healing nanocomposite hydrogels based on nano clay (laponite), multiwalled carbon nanotubes (CNTs), and N-isopropyl acrylamide, shown in Figure 9d [234]. The presented nanocomposite hydrogels displayed good electrical conductivity, rapid self-healing and adhesive properties, flexible and stretchable mechanical properties, and high sensitivity to near-infrared light and temperature [234]. Courbat reported hybrid resistive temperature and capacitive humidity sensors of silver nanoparticles on paper designed by inkjet printing, which foresees the possibility of low-cost environmental sensors [235]. Li et al. demonstrated the 3D printing of highly stretchable coplanar sensors for tactile and electrochemical sensing applications. Interdigital electrodes based on CNT-PDMS composites were printed for strain and tactile sensing applications [236]. The fabricated double-vortex sensors based on CNT-Ecoflex for highly sensitive electrochemical sensing applications have achieved the detection limit as low as 1 μM for NaCl aqueous solution [236]. This kind of stretchable tactile and electrochemical sensors hold great promise in future integrated wearable devices. A recent device presented by inkjet printing can simultaneously measure relative humidity, temperature, compressive and tensile bending. The hybrid sensor can be used as smart packaging-labeling and disposable biosensors [237].

**Figure 9 micromachines-13-00642-f009:**
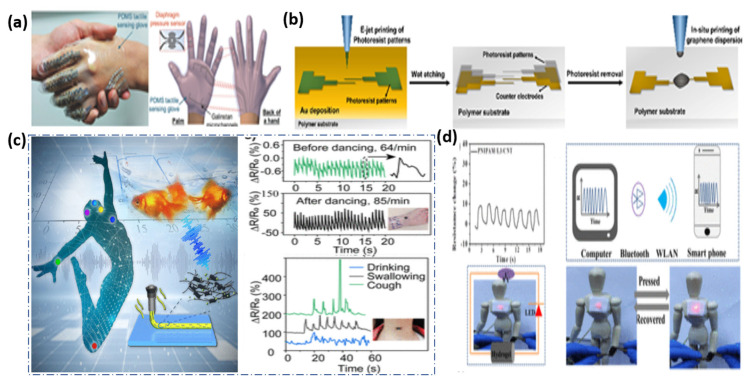
(**a**) Photograph of hand-shaking wearing the PDMS tactile sensing glove (left); Schematic of the PDMS tactile sensing glove (right) [218]. (**b**) Schematic of the process for fabricating fully printed graphene photodetector devices using the mask-free direct-writing method [214]. (**c**) flexible composite pressure sensors for underwater monitoring (left); pulse wave curves and response signals under different conditions (right [151]). (**d**) Photographs of the increased light intensity (circuit voltage was 3 V) after NIR light exposure (top); Pressure-dependent conductivity of PNIPAM/ Laponite/CNT hydrogels (bottom) [234].

**Table 3 micromachines-13-00642-t003:** 3DP electronics.

3DP Types	Sensing Materials	Types of Sensors	Application	Ref.
EHDP	Molten metal ink	Touch sensor	Flexible/stretchable devices	[65]
EHDP	Ag nanoink/ITO	Capacitive touch sensors	Flexible displays	[203]
EHDP	Molten polymer	Microcantilever sensor	Detecting multiple analytes	[207]
EHDP	PEDOT:PSS/NMP PEDOT:PSS/PVP/NMP PEDOT:PSS/PVP/Nafion/NMP	Pressure/strain sensor	Flexible robotic skin	[143]
EHDP	PEO/PANI/G	Piezoresistive sensor	Healthcare Environmental/bio-related monitoring	[151]
Inkjet P	TPU/CB	Piezoresistive sensor	Health monitoring Robotics tactile sensing Human machine interfaces	[66]
Inkjet P	Elastomer/pencil	Capacitive sensor	Touchpad Human-interface machine	[211]
Inkjet P	PET/Mylar/Ag NPs	Capacitive acoustic resonators	Navigation of drones	[205]
EHDP	PEDOT:PSS/poly(3-hexylthiophene-2,5-diyl) (P3HT)	Ion-gel transistor	Logic circuit Display backbone	[145]
EHDP	PEDOT:PSS	OTFT	Wearable sensor	[148]
EHDP	PEDOT:PSS/CNT	OTFT	Wearable sensor	[149]
EHDP	PVDF/BaTiO_3_	Piezoelectric sensor	Gait analysis	[212]
Inkjet P	G/tungsten disulfide (WS_2_)/Si/SiO_2_	Photosensor	Optical communications	[213]
Inkjet P	PVP/parylene-C/Ag nano ink	OTFT	Wearable sensor	[208]
EHDP	G ink	Photodetector	Optical communications	[214]
/(print)	G/WSe_2_/boron nitride (BN)	TFT	Optical communications	[209]
/(3DP)	Polylactic acid (PLA)/MWCNT	Liquid sensor	Substance detection	[220]
EHDP	MoS_2_	Gas sensor	NO_2_/NH_3_ detection	[223]
Extrusion P	CNTs/PLA	Liquid sensor	Smart sensors in textile	[222]
Soft lithography assisted 3DP	Colorless resin	Microfluidics sensor	Analysis of nitrate in tap water	[221]
/(3DP)	G/PLA	Glucose biosensor	Glucose detection	[230]
/(3DP)	MXene/PEDOT: PSS	Electrochemical sensor	Nucleic acid detection	[228]
/(3DP)	MXene quantum dot-/liposome	Electrochemical sensor	Antibody detection	[229]
/	Reduced graphene oxide foam (rGOF)	Pressure/temperature sensor	Electronic skin	[232]
/(3DP)	PNIPAM/Laponite/CNT	Pressure/near-infrared light/temperature sensor	Human motion sensing Stimuli-responsive Electrical devices Wearable electronics	[234]
Inkjet P	Ag NPs	Resistive temperature	Environmental sensor	[235]
/(3DP)	CNT-ecoflex CNT-PDMS	capacitive/electrochemical sensor	Stretchable tactile/Electrochemical sensors	[236]
Inkjet P	Ag/PEDOT: PSS	humidity/temperature/compressive/strain sensor	Disposable biosensor Smart Packaging-labeling	[237]

## 5. Conclusions

The 3D printing scale ability has advanced tremendously to develop many biomedical and electronic devices on the micro and nano scales. Many challenges still prevent the progress and innovation of small scale devices and continue to leave a gap in research for developing novelties applicable to biomedical and electronic devices. The printing techniques outlined in this review demonstrate that the technique of choice should be determined based on the application outcome, as some techniques have challenges and capabilities different than others. Two photon polymerization has difficulties in mass production due to its low throughput; however, it is advantageous due to its high spatial resolution and variety of printing material. Dip pen nanolithography is limited to only allowing certain compounds to disperse, but its technique can produce micro and nano structures through patterns by changing the AFM tip. Thermal inkjet printing precision is unfortunately limited to the tip diameter, although the technique is capable of using a high ink content for printing which results in high printing efficiency. Piezoelectric inkjet printing can suffer from issues associated with clogging of the orifice during downtime which may result in unstable or non-existent ejection. Nevertheless, the printing principle is still beneficial as it has high aspect ratios within the micrometer scale, meanwhile using minimal material possible. These printing technologies have extensive applications covering drug delivery systems, scaffolds for tissue engineering and the fabrication of electronic devices. The scalability is still being reduced in the 3D printing field with printing technology improving; there is no doubt that the innovation possibilities are endless and the micro/nano scale will continue to be further incorporated in the evolution of biomedical and electronic devices. 

## Figures and Tables

**Figure 1 micromachines-13-00642-f001:**
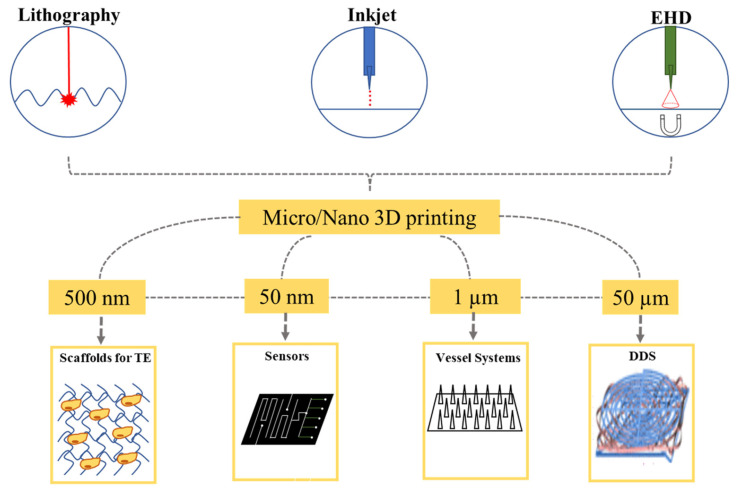
Overview of micro/nano scale printing processes and applications that have been 3D printed from 500 nm to 50 µm via lithography, inkjet and EHD printing [18].

**Figure 2 micromachines-13-00642-f002:**
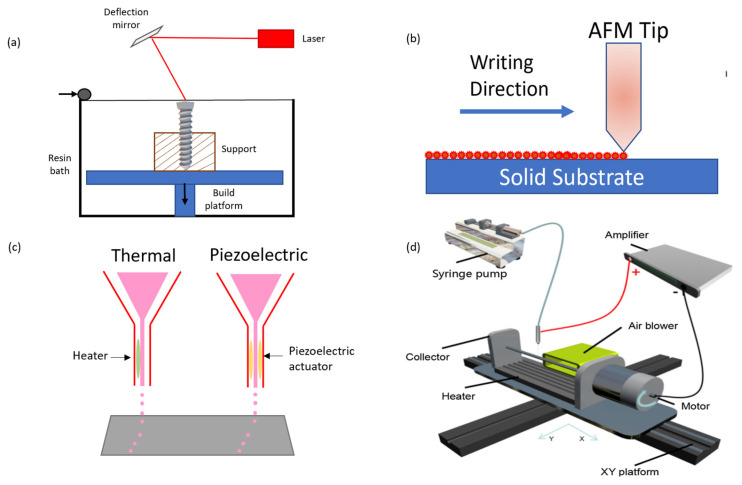
Illustrations of various 3D printing processes showing (**a**) stereolithography, (**b**) dip pen nanolithography (**c**) Inkjet printing and (**d**) EHD printing [22].

**Figure 3 micromachines-13-00642-f003:**
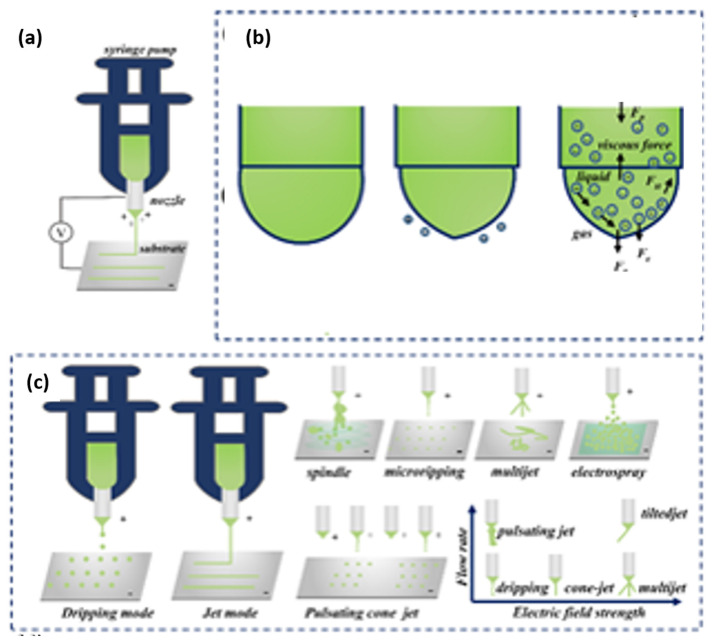
EHD printing principle and modes. (**a**) EHDP printing system. (**b**) Cone formation and force analysis. (**c**) EHD jet modes.

**Figure 4 micromachines-13-00642-f004:**
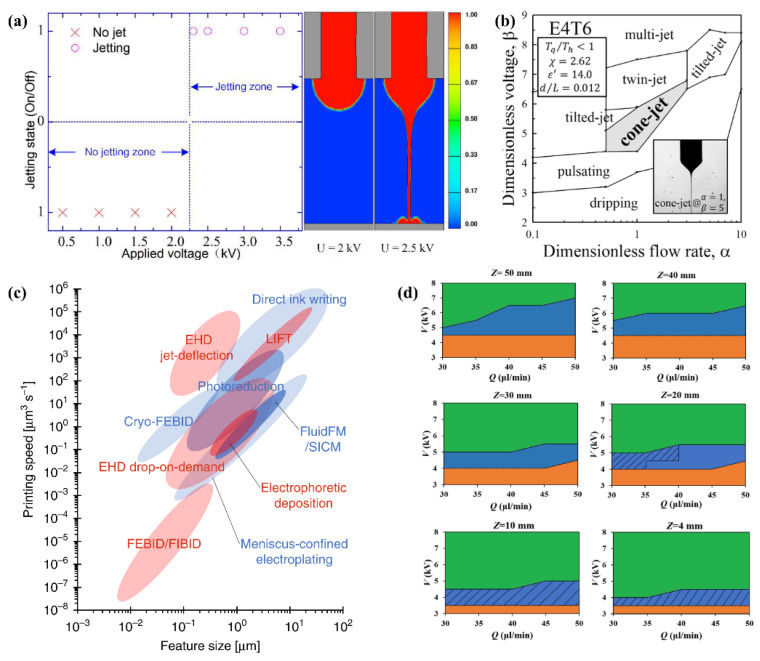
(**a**) Relationship between EHD jetting state and applied voltage (colorful images are meniscus shape at the same calculating moment (t = 0.4 ms) in 2.0 kV and 2.5 kV, respectively) [86]. (**b**) EHD jetting states change with the variations of applied voltage and flow rate when other parameters are certain values [94]. (E4T6, dimensionless velocity (χ) = 2.62, permittivity (ε′) = 14.0, d/L is the ratio of nozzle diameter (**d**) and nozzle–counter electrode distance (L), Tq/Th is the ratio of two characteristic times that determine the jetting system). (**c**) Map of printing capabilities in terms of printing speed and feature size for EHDP technologies providing submicron resolution [95]. (**d**) EHD jet modes at different flow rate Q, working distance Z, and applied voltage V [96].

**Figure 5 micromachines-13-00642-f005:**
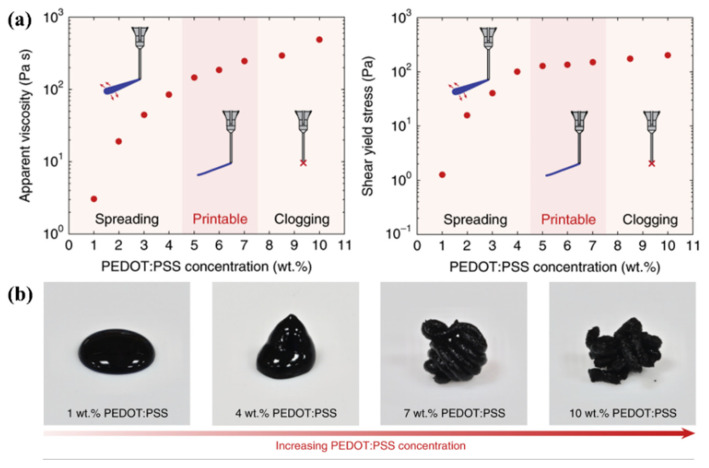
(**a**) Viscosity of PEDOT: PSS inks as a function of PEDOT: PSS nanofibril concentration (left); Shear yield stress of inks as a function of PEDOT: PSS nanofibril concentration (right) [104]. (**b**) Images of re-dispersed suspensions with varying PEDOT: PSS nanofibril concentration [104].

**Figure 6 micromachines-13-00642-f006:**
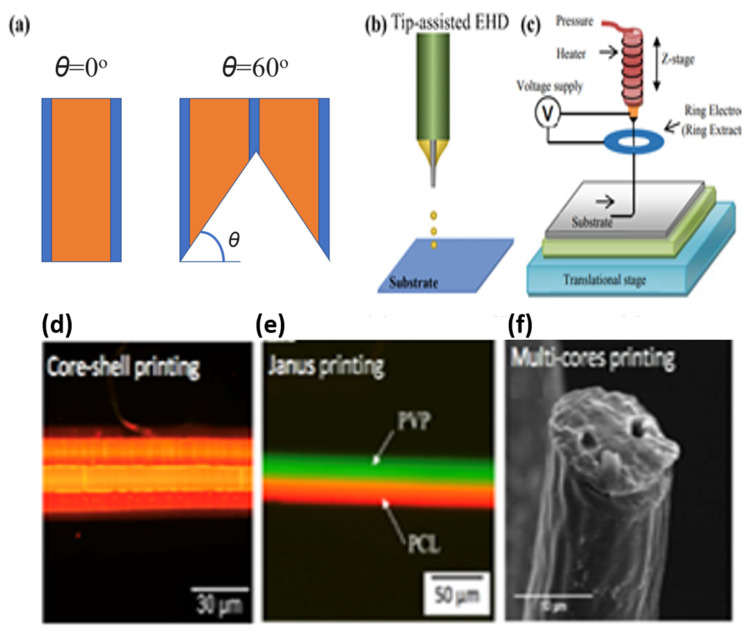
(**a**) Flat needle and angular needle (θ = 60°) [107]. (**b**) Tip-assisted EHD [115]. (**c**) EHDP with the added metal ring [116]. (**d**) core-shell printing [123]. (**e**) Janus printing [88]. (**f**) Multi-cores printing [124].

**Figure 8 micromachines-13-00642-f008:**
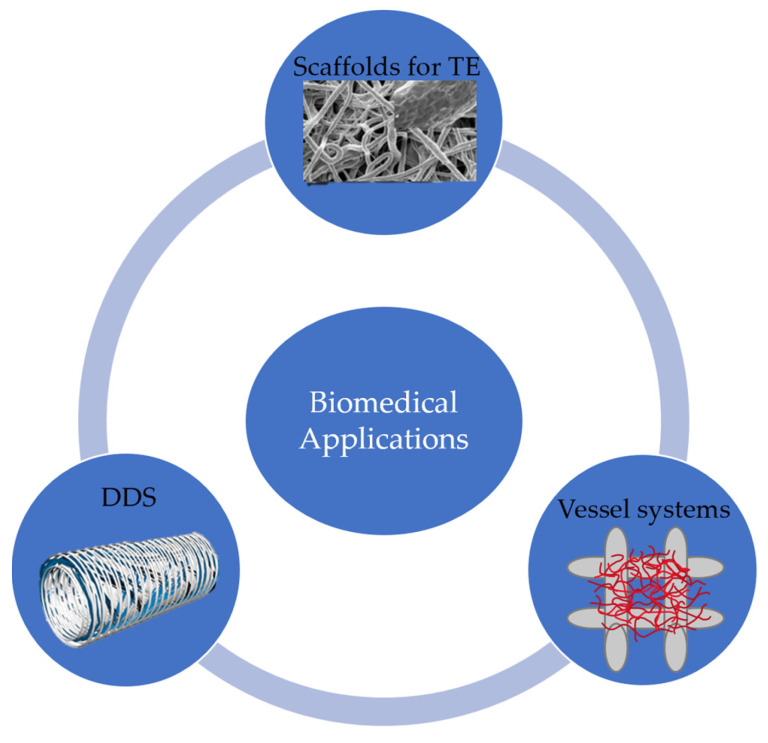
Biomedical applications of micro/nano 3D printed devices [8,176].

**Table 1 micromachines-13-00642-t001:** Advantages and disadvantages of DPN technology.

Advantages	Disadvantages
Nano-patterns do not require optical apparatus [36]	Low throughput for batch manufacturing [37,38]
Deposited material is controlled by hydrophobicity of the surface [36]	Probe tip may be subjected to wear resulting in poor reproducibility [36]
AFM tip can be changed to generate a random pattern [36]	Ambient conditions need to be constant as humidity affects printing ink ensuing in deformed pattern. [36]
In situ imaging capability [33]	Hollow AFM tip can only permit certain compounds to pass. [36]

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
