# Peer review of "High Precision 3D Printing for Micro to Nano Scale Biomedical and Electronic Devices"

_micromachines, 2022, doi:10.3390/mi13040642_

Round 1

Reviewer 1 Report

This is an interesting review, the authors conducted a detailed literature review on“High Precision 3D Printing for Micro to Nano Scale Biomedical 2 and Electronic Devices”. Micro and nano scale printing has encouraged innovation in the aforementioned sectors, due to the ability to control design, material and chemical properties at a highly precise level, which is advantageous in creating a high surface area to volume ratio and altering the overall products’ mechanical and physical properties. But there are still some details that need to be worked out, which could make this review even better.

1.Microstructural control of 3D printing is a very important aspect, especially for the fabrication of porous scaffolds. But it is not stated in this review. Here are some references. Please add and discuss.

  • A multifunctional micropore-forming bioink with enhanced anti-bacterial and anti-inflammatory properties. M Wang, W Li, Z Luo, G Tang, X Mu, X Kuang, J Guo, Z Zhao, RS Flores, Biofabrication
  • Aubert, T., Huang, JY., Ma, K. et al. Porous cage-derived nanomaterial inks for direct and internal three-dimensional printing. Nat Commun 11, 4695 (2020). https://doi.org/10.1038/s41467-020-18495-5
  • Vertical Extrusion Cryo (bio) printing for Anisotropic Tissue Manufacturing. Z Luo, G Tang, H Ravanbakhsh, W Li, M Wang, X Kuang, ADVANCED MATERIALS 2022
  • Freeform cell-laden cryobioprinting for shelf-ready tissue fabrication and storage. H Ravanbakhsh, Z Luo, X Zhang, S Maharjan, HS Mirkarimi, G Tang, ...Matter 5 (2), 573-593
  • Dong, Z., Cui, H., Zhang, H. et al. 3D printing of inherently nanoporous polymers via polymerization-induced phase separation. Nat Commun 12, 247 (2021). https://doi.org/10.1038/s41467-020-20498-1
  • Freeze-Casting with 3D-Printed Templates Creates Anisotropic Microchannels and Patterned Macrochannels within Biomimetic Nanofiber Aerogels for Rapid Cellular Infiltration;Johnson V. John, Alec McCarthy, Hongjun Wang, Zeyu Luo, Hongbin Li, Zixuan Wang, Feng Cheng, Yu Shrike Zhang, Jingwei Xie
  1. Nano printing is a quite important aspect of the high precise 3D printing. There are several high impact nano scale 3D printing papers that should be cited and have a discussion.

  • Hahn, V., Messer, T., Bojanowski, N.M. et al. Two-step absorption instead of two-photon absorption in 3D nanoprinting. Nat. Photon. 15, 932–938 (2021). https://doi.org/10.1038/s41566-021-00906-8
  • Coelho, S., Baek, J., Walsh, J. et al. Direct-laser writing for subnanometer focusing and single-molecule imaging. Nat Commun 13, 647 (2022). https://doi.org/10.1038/s41467-022-28219-6
  • Wen, X., Zhang, B., Wang, W. et al. 3D-printed silica with nanoscale resolution. Nat. Mater. 20, 1506–1511 (2021). https://doi.org/10.1038/s41563-021-01111-2

  1. Multi-material printing is also an important research direction for precise 3D printing. There are many ways to achieve 3D printing of various materials. There are several related literatures here, please add and discuss.
  • Digital light processing based bioprinting with composable gradients. M Wang, W Li, LS Mille, T Ching, Z Luo, G Tang, CE Garciamendez, Advanced Materials 34 (1), 2107038
  • Uniaxial and Coaxial Vertical Embedded Extrusion Bioprinting. L Lian, C Zhou, G Tang, M Xie, Z Wang, Z Luo, J Japo, D Wang, J Zhou,Advanced Healthcare Materials, 2102411
  • Sinha, R., Cámara-Torres, M., Scopece, P. et al. A hybrid additive manufacturing platform to create bulk and surface composition gradients on scaffolds for tissue regeneration. Nat Commun 12, 500 (2021). https://doi.org/10.1038/s41467-020-20865-y

  1. for the inject printing part, there are several quite fancy papers should be cited.
  • Zheng, F., Wang, Z., Huang, J. et al. Inkjet printing-based fabrication of microscale 3D ice structures. Microsyst Nanoeng 6, 89 (2020). https://doi.org/10.1038/s41378-020-00199-
  • Jung, W., Jung, YH., Pikhitsa, P.V. et al. Three-dimensional nanoprinting via charged aerosol jets. Nature 592, 54–59 (2021). https://doi.org/10.1038/s41586-021-03353-1

Author Response

Reviewer 1

This is an interesting review, the authors conducted a detailed literature review on“High Precision 3D Printing for Micro to Nano Scale Biomedical 2 and Electronic Devices”. Micro and nano scale printing has encouraged innovation in the aforementioned sectors, due to the ability to control design, material and chemical properties at a highly precise level, which is advantageous in creating a high surface area to volume ratio and altering the overall products’ mechanical and physical properties. But there are still some details that need to be worked out, which could make this review even better.

  1. Microstructural control of 3D printing is a very important aspect, especially for the fabrication of porous scaffolds. But it is not stated in this review. Here are some references. Please add and discuss.
  • A multifunctional micropore-forming bioink with enhanced anti-bacterial and anti-inflammatory properties. M Wang, W Li, Z Luo, G Tang, X Mu, X Kuang, J Guo, Z Zhao, RS Flores, Biofabrication
  • Aubert, T., Huang, JY., Ma, K. et al. Porous cage-derived nanomaterial inks for direct and internal three-dimensional printing. Nat Commun 11, 4695 (2020). https://doi.org/10.1038/s41467-020-18495-5
  • Vertical Extrusion Cryo (bio) printing for Anisotropic Tissue Manufacturing. Z Luo, G Tang, H Ravanbakhsh, W Li, M Wang, X Kuang, ADVANCED MATERIALS 2022
  • Freeform cell-laden cryobioprinting for shelf-ready tissue fabrication and storage. H Ravanbakhsh, Z Luo, X Zhang, S Maharjan, HS Mirkarimi, G Tang, ...Matter 5 (2), 573-593
  • Dong, Z., Cui, H., Zhang, H. et al. 3D printing of inherently nanoporous polymers via polymerization-induced phase separation. Nat Commun 12, 247 (2021). https://doi.org/10.1038/s41467-020-20498-1
  • Freeze-Casting with 3D-Printed Templates Creates Anisotropic Microchannels and Patterned Macrochannels within Biomimetic Nanofiber Aerogels for Rapid Cellular Infiltration;Johnson V. John, Alec McCarthy, Hongjun Wang, Zeyu Luo, Hongbin Li, Zixuan Wang, Feng Cheng, Yu Shrike Zhang, Jingwei Xie

Thank you for your suggestion and associated references. Although the current review focuses on high precision printing of micro to nano scale biomedical and electronic devices, the discussion on porous scaffolds and its importance in 3D fabrication has been added to page 23, section 4.1.2. Bioink related references have been inserted on page 25, section 4.1.3.

The fabrication of three-dimensional porous architectures are favorable in the design of biomedical devices as the modified surface is ideal for enhancing cell activity and infiltration in tissue repair and regeneration. [1] Wang et al. successfully introduced a porous structure via VAT polymerization, extrusion and handheld bioprinting by using a micropore-forming bioink. Dextran was used as the porogen to generate a microporous structure measuring 43.7 ± 7.5 µm. The microstructure proved to be superior in cellular behavior in comparison to non-microporous structures.[2] [3]Porous structures portray an interconnected network of material which can alter the transport pathways within a printed structure.[4] Aubert et al. reported novelty digital light processing of mesoporous multi- materials which avoids the calcification step that is required in the extrusion printing process of mesoporous structures. The methodology is additionally advantageous as the direct printing of mesopores enabled pore accessibility and control over the internal structure.[4] The challenge remains in producing sub- micrometer scale features as part of a larger polymer-based design, however combined digital light processing with polymerization induced phase separation to manufacture macroscopic polymer objects with a controlled inherent nano porosity is an alternative method for biomedical applications.[5]

An existing challenge in relation to bioinks is the poor shelf availability as a result of fabrication and storage complications.[6]

  1. Nano printing is a quite important aspect of the high precise 3D printing. There are several high impact nano scale 3D printing papers that should be cited and have a discussion.

  • Hahn, V., Messer, T., Bojanowski, N.M. et al. Two-step absorption instead of two-photon absorption in 3D nanoprinting. Nat. Photon. 15, 932–938 (2021). https://doi.org/10.1038/s41566-021-00906-8
  • Coelho, S., Baek, J., Walsh, J. et al. Direct-laser writing for subnanometer focusing and single-molecule imaging. Nat Commun 13, 647 (2022). https://doi.org/10.1038/s41467-022-28219-6
  • Wen, X., Zhang, B., Wang, W. et al. 3D-printed silica with nanoscale resolution. Nat. Mater. 20, 1506–1511 (2021). https://doi.org/10.1038/s41563-021-01111-2

Thank you for your suggestion, sections below on nanoscale printing references to be added to page 18, section 3.3 and page 4, section 2.1.1. respectively:

The printing ink is a key factor in achieving micro and nano scale features in a 3D printed design. By using a printing ink containing nanoparticles measuring around 10 nm, nanostructures can be achieved. Wen et al. produced nanostructures using silica nanoparticles to a resolution of 200 nm which demonstrated attractive optical properties.[7]

Two step absorption is preferred to the previous technique of two photon absorption in the TPP printing process as the latter mechanism, although is capable of nanofabrication[8], it is associated with issues around cost, reliability and higher order processes. The absorption method possesses that same quadratic optical nonlinearity while offering miniaturization and cost reduction of 3D nano scale printers. [9]

  1. Multi-material printing is also an important research direction for precise 3D printing. There are many ways to achieve 3D printing of various materials. There are several related literatures here, please add and discuss.

  • Digital light processing based bioprinting with composable gradients. M Wang, W Li, LS Mille, T Ching, Z Luo, G Tang, CE Garciamendez, Advanced Materials 34 (1), 2107038
  • Uniaxial and Coaxial Vertical Embedded Extrusion Bioprinting. L Lian, C Zhou, G Tang, M Xie, Z Wang, Z Luo, J Japo, D Wang, J Zhou,Advanced Healthcare Materials, 2102411
  • Sinha, R., Cámara-Torres, M., Scopece, P. et al. A hybrid additive manufacturing platform to create bulk and surface composition gradients on scaffolds for tissue regeneration. Nat Commun 12, 500 (2021). https://doi.org/10.1038/s41467-020-20865-y

Thank you, further detail on multi-material printing to be added to page 23, section 4.1.2, text detailed below:

Biomimicry is an asset desired in biomedical engineering, particularly in a 3D manufactured scaffold for tissue engineering. It is understandable that a biomedical device constructed from a single material will not meet all the requirements to succeed in a biological environment. Therefore by fabricating a biomedical device using multiple materials, the chemical and physical properties of the overall product will be an advancement upon the single material design. Multi-material design can influence factors of the microenvironment and is desired in establishing the functionality of engineering tissues.[10]

Continuous composition gradients are desirable in scaffolds for tissue engineering which can be achieved via multiple material printing ink. This requires material dosing and mixing controls to modify surfaces to promote cell adhesion. Sinha et al. conducted this technique using a dual material print head to produce continuous composition gradients of thermoplastic scaffolds with improved mechanical properties and enhanced cell adhesion.[11]

  1. for the inkjet printing part, there are several quite fancy papers should be cited.
  • Zheng, F., Wang, Z., Huang, J. et al. Inkjet printing-based fabrication of microscale 3D ice structures. Microsyst Nanoeng 6, 89 (2020). https://doi.org/10.1038/s41378-020-00199-
  • Jung, W., Jung, YH., Pikhitsa, P.V. et al. Three-dimensional nanoprinting via charged aerosol jets. Nature 592, 54–59 (2021). https://doi.org/10.1038/s41586-021-03353-1

The papers are to be cited and discussed as below: (page 5)

Microfabrication by inkjet printing has been often require expensive equipment and a supporting or protecting layer. However, Zheng et al. proposed an economical method based on ice printing which can form structures with a maximum height of 2000 µm. The innovative method does not require additional support or removing processes and the need to introduce additional chemicals. The ice printing inkjet technique is promising as it has great potential in producing porous scaffolds of salt metal nanoparticles. [12] The geometric ability on inkjet printing can be controlled by moving the substrate during the printing process, Jung et al. demonstrated this by achieving nanopillars with a feature size of 85 nm.

A correlated challenge to inkjet printing lies with the limitation in geometries that can be achieved using high purity metals. However Jung et al. successfully conducted experiments that produced 3D arrays of metal nanostructures with flexible geometry with nanoscale features as small as a few hundred nanometers.[13]

Reviewer 2 Report

This is a comprehensive, accurate and updated review on a hot topic of modern science and technology. The paper is highly suitable for the Journal and I expect that it will be of great interest to the Journal readership.

I have no hesitation to recommend publication; I just recommend a couple of improvements listed below.

  1. The manuscript is generally well written but some typos need to be corrected; furthermore, check the subscripts in the formulas of printing materials in section 3.3 (e.g. SiO2, Al2O3 etc.).

  1. Inks may have pretty different characteristics depending on the material used – e.g. polymeric or ceramic ink. In the latter case (printing of ceramic suspensions), some key parameters could be further discussed shortly, e.g. ceramic particle size and distribution on ink rheology etc.

  1. Again, the part on 3DP applied to bioceramics and bioglasses could be expanded a bit; you could cite for instance this couple of papers:

Additive manufacturing of bioactive glasses and silicate bioceramics, J. Ceram. Sci. Technol. 6 (2015) 75–86.

3D printing of hierarchical scaffolds based on mesoporous bioactive glasses (MBGs)-fundamentals and applications. Materials 2020;13:1688.

  1. A short section addressed to a market perspective would further strengthen the paper.

Author Response

Reviewer 2

This is a comprehensive, accurate and updated review on a hot topic of modern science and technology. The paper is highly suitable for the Journal and I expect that it will be of great interest to the Journal readership.

I have no hesitation to recommend publication; I just recommend a couple of improvements listed below.

  1. The manuscript is generally well written but some typos need to be corrected; furthermore, check the subscripts in the formulas of printing materials in section 3.3 (e.g. SiO2, Al2O3 etc.).

Thank you, typos and formula subscripts have been corrected.

  1. Inks may have pretty different characteristics depending on the material used – e.g. polymeric or ceramic ink. In the latter case (printing of ceramic suspensions), some key parameters could be further discussed shortly, e.g. ceramic particle size and distribution on ink rheology etc.

The following section to be added to page 17, section 3.3

The particle size of the printing ink in many 3D printing processes is a considerable factor in the accuracy of the resultant print. For example, in photopolymerization larger particle sizes reduces the rate of polymerization as a result of light scattering and absorption. Additionally in inkjet printing the particle size needs to be small enough to prevent clogging or blockages in the printing nozzle. However, if the particles are too small, agglomeration can occur. Therefore when printing with ceramic based ink, it must be homogenously dispersed to pass through the printing head with ease.[14] The solid content of ceramic inks for additive manufacturing can affect the materials rheological behavior.[15] It is worthwhile conducting rheological assessments of the material to determine the extrudability and shape fidelity.[16] Printing requirements that are a factor of rheological properties is the ability to retain the shape of the printing nozzle, be self-supporting and accurately reproduce the printing path. These conditions can be satisfied when the ink behaves as a fluid during printing and presents elastic behavior at rest.[16]

  1. Again, the part on 3DP applied to bioceramics and bioglasses could be expanded a bit; you could cite for instance this couple of papers:

Additive manufacturing of bioactive glasses and silicate bioceramics, J. Ceram. Sci. Technol. 6 (2015) 75–86.

3D printing of hierarchical scaffolds based on mesoporous bioactive glasses (MBGs)-fundamentals and applications. Materials 2020;13:1688.

We appreciate your recommended citations, the proposed text below is to be added to page 18, section 3.3

Additive manufacturing using printing inks that are suitable biomaterials has become an increasingly popular area within research and the development of biomedical devices. Bioglasses and Bioceramics are examples of biocompatible materials used as printing inks to form a patient specific design with tailored properties to serve their purpose in a biological environment. Bioglass and bioceramics offer advantages over Ca-P based materials for bone regeneration due to their high bioactivity, angiogenic properties and osteoinduction capabilities. These bioactive materials have the ability to induce antibacterial effects and cellular responses.[17] Micro and nano structures are necessary for biomedical applications, as they are required to truly mimic the microenvironment. Hence mesoporous active glasses has gained significant interest in science as proposed drug delivery systems due to the mesoporous structure offering control on release kinetics, and therefore offering a confined therapeutic treatment.[18]

  1. A short section addressed to a market perspective would further strengthen the paper.

The market perspective to be expanded upon following market value in the introduction as written below:

The sales market of 3D printers seen 2.2 million shipments in 2021, which is expected to increase to 21.5 million by 2030. This increase is estimated to be a result of research and development, particularly within the medical and automotive sectors. Prototyping is the leading application in the 3D printing market, attributing to more than 55% of global revenue in 2021. This is likely due to prototyping achieving higher accuracy to develop high end products.[19]